# RAGE is a key regulator of ductular reaction-mediated fibrosis during cholestasis

Wai-Ling Macrina Lam[1,2], Gisela Gabernet[3], Tanja Poth[4], Melanie Sator-Schmitt[1], Morgana Barroso Oquendo [ID][3], Bettina Kast [ID][1], Sabrina Lohr[1], Aurora de Ponti[1], Lena Weiß[1], Martin Schneider [ID][5], Dominic Helm [ID][5], Karin Müller-Decker[6], Peter Schirmacher[4], Mathias Heikenwälder[7], Ursula Klingmüller[8], Doris Schneller[1], Fabian Geisler[9], Sven Nahnsen[3] & Peter Angel [ID][1]✉

## Abstract

**Ductular reaction (DR) is the hallmark of cholestatic diseases manifested in the proliferation of bile ductules lined by biliary epithelial cells (BECs). It is commonly associated with an increased risk of fibrosis and liver failure. The receptor for advanced glycation end products (RAGE) was identified as a critical mediator of DR during chronic injury. Yet, the direct link between RAGE-mediated DR and fibrosis as well as the mode of interaction between BECs and hepatic stellate cells (HSCs) to drive fibrosis remain elusive. Here, we delineate the specific function of RAGE on BECs during DR and its potential association with fibrosis in the context of cholestasis. Employing a biliary lineage tracing cholestatic liver injury mouse model, combined with whole transcriptome sequencing and in vitro analyses, we reveal a role for BEC-specific *Rage* activity in fostering a pro-fibrotic milieu. *RAGE* is predominantly expressed in BECs and contributes to DR. Notch ligand Jagged1 is secreted from activated BECs in a *Rage*-dependent manner and signals HSCs in trans, eventually enhancing fibrosis during cholestasis.**

**Keywords** Biliary Epithelial Cells; Chronic Liver Injury; Genetically Modified Mice; Hepatic Stellate Cells; Receptor for Advanced Glycation End Products
**Subject Categories** Molecular Biology of Disease; Signal Transduction

## Introduction

Hepatic fibrosis is implicated in most chronic liver diseases. Fibrosis results from two general types of persistent hepatic injury: hepatotoxic and cholestatic injury. Hepatotoxic injury generally arises from viral infections, steatohepatitis or metabolic syndrome, such as nonalcoholic steatohepatitis, whereas cholestatic injury is commonly caused by the obstruction of bile flow due to bile duct paucity or inflammation within the biliary system (Kisseleva and Brenner, 2021). During cholestasis, hepatic damage as a consequence of the accumulation of bile acids may lead to progressive liver diseases and potentially provoke liver failure (Turnpenny and Ellard, 2012). Fibrosis is characterized by the scarring nature of tissues due to myofibroblastic hepatic stellate cell (HSC)-derived excessive deposition of extracellular matrix. Activated portal fibroblasts and HSCs were well-recognized to be the major source that contribute to fibrosis, as demonstrated e.g., in experimental cholestatic-associated *Mdr2*-knockout and bile duct ligation mouse models (Iwaisako et al, 2014; Nishio et al, 2019).

The progression and the degree of fibrosis correlates with ductular reaction (DR) in most of the etiologies of human liver diseases (Sato et al, 2019; Lowes et al, 1999; Sancho-Bru et al, 2012; Jung et al, 2008; Nobili et al, 2012). DR is histologically characterized by the formation of tubular bile duct structure that is lined by the biliary epithelial cells (BECs). BECs comprise only 3–5% of total liver. They represent a population of quiescent liver resident cells residing at the biliary compartment of the periportal area. Extensive lineage tracing studies have employed various murine disease models to investigate the pathophysiological role of BECs. It appears that BECs encompass a dual role during chronic injury depending on the modulation of signaling pathways in different context of local tissue microenvironment. Numerous studies reported that BECs contribute to liver regeneration upon injury (Furuyama et al 2011; Shin et al, 2015; Huch et al, 2013; Español–Suñer et al, 2012); on the contrary, it was also evident that BECs in a currently ill-defined fashion contribute to fibrosis (Kuramitsu et al, 2013; Clouston et al, 2005) and formation of HCC (Lee et al, 2006; Tummala et al, 2017; Holczbauer et al, 2013). Recent advanced multidimensional imaging study has shed light on

[1]Division of Signal Transduction and Growth Control, German Cancer Research Center (DKFZ), DKFZ-ZMBH Alliance, Heidelberg, Germany. [2]Faculty of Biosciences, Ruprecht Karl University of Heidelberg, Heidelberg, Germany. [3]Quantitative Biology Center (QBiC), Eberhard Karls University of Tübingen, Tübingen, Germany. [4]Institute of Pathology, University Hospital Heidelberg, Heidelberg, Germany. [5]Protein Analysis Unit, Genomics and Proteomics Core Facility, German Cancer Research Center (DKFZ), Heidelberg, Germany. [6]Tumor Models Unit, Center for Preclinical Research, German Cancer Research Center (DKFZ), Heidelberg, Germany. [7]Division of Chronic Inflammation and Cancer, German Cancer Research Center (DKFZ), Heidelberg, Germany. [8]Division of Systems Biology of Signal Transduction, German Cancer Research Center (DKFZ), Heidelberg, Germany. [9]TUM School of Medicine and Health, Department of Clinical Medicine – Clinical Department for Internal Medicine II, University Medical Center, Technical University of Munich, München, Germany. ✉E-mail: p.angel@dkfz-heidelberg.de

the fundamental role of BECs, and revealed that these BEC-lined, duct-like structure functions as an escape route for accumulative bile during cholestasis (Kamimoto et al, 2020). Our previous data revealed that DR and the onset of fibrosis and hepatocellular carcinoma formation was modulated by the receptor for advanced glycation end products (RAGE) in the Choline-deficient Ethionine-supplemented (CDE) diet-induced injury model and the *Mdr2*-knockout genetic model of cholestasis (Pusterla et al, 2013), suggesting that RAGE is a regulator for DR-associated fibrosis.

RAGE belongs to the immunoglobulin superfamily of cell surface receptor. It functions as a pattern recognition receptor that engages a broad repertoire of ligands, including advance glycation end products (AGEs), amyloid-β peptide and damage-associated molecular pattern (DAMP) molecules, including HMGB1 and S100 proteins (Bierhaus et al, 2005). Under physiological conditions, RAGE is expressed ubiquitously at low basal level in all types of tissues, except alveolar epithelial cells in the lung, which exhibit a significantly higher abundance of RAGE molecules. Under conditions of chronic tissue damage and inflammation, RAGE expression increases significantly in immune cell populations (Weinhage et al, 2020, Kierdorf and Fritz, 2013). RAGE signaling plays an essential role in modulating the tissue microenvironment and its activation is required for the perpetuation of inflammatory responses (Weinhage et al, 2020; Kierdorf and Fritz, 2013). Nevertheless, neither the cell-intrinsic function of RAGE as the main direct driver for DR, nor the specific cell type(s) that are responsible for RAGE-dependent effects in the context of cholestatic fibrosis are clearly defined. Thus, elucidating the manifestation of RAGE specifically in the cholestatic tissue microenvironment will enhance our understanding of the underlying mechanism of cholestasis-associated fibrosis.

In this study, we have utilized the well-established murine CDE diet-induced injury model that resembles DR in human chronic inflammation-associated liver diseases (Akhurst, 2001; Passman et al, 2015; Clerbaux et al, 2018). We show that RAGE activity in BECs is indispensable for DR formation and DR-mediated fibrosis during cholestasis. Furthermore, we have unraveled the RAGE-dependent interplay between BECs and HSCs on the molecular level. Direct and indirect co-cultures of BECs and HSCs, combined with genome-wide expression analysis and mass spectrometry show that BEC-derived Notch ligand Jagged1 induces HSC activation. These results imply that RAGE represents a novel vulnerability for the treatment of cholestasis-associated fibrosis.

## Results

### RAGE is enriched in activated BECs during hepatic injury

RAGE is a main regulator that mediates BEC expansion, onset of liver fibrosis and tumor formation in a chronic inflammation-associated HCC model either in a cell-autonomous, or via a more indirect manner (Pusterla et al, 2013). BECs line the extra- and intrahepatic biliary tree. Under homeostatic conditions, A6/CK19+ bile ducts/ductules are confined to the portal space in direct proximity to the portal vein (Appendix Fig. S1A, upper panel). To induce chronic insult in the mouse liver for subsequent functional characterization of BECs, we administered the well-established ad libitum CDE diet to C57BL/6 mice for 3 weeks. Under these conditions, the population of BECs labeled by the BEC-specific

markers A6 and Cytokeratin-19 (CK19) A6$^+$/CK19$^+$ ductules (DR) is not confined to the portal area anymore but is additionally prominently visible within the liver parenchyma (A6$^+$/CK19$^+$ ductules (DR); Appendix Fig. S1A, lower panels). Importantly, the expression of *Rage* in isolated BECs is 130-fold higher than the level measured in purified primary hepatocytes (Appendix Fig. S1B).

### RAGE in BECs is required for mounting a DR phenotype

To specifically define the role of RAGE in BECs in establishing DR in the context of chronic liver injury we generated a tamoxifen-inducible BEC-specific conditional Rage knockout model by crossing the $R26^{Tom}$ *Hnf1b-CreER* mouse strain (Jörs et al, 2015) with RAGE/EGFP reporter mice described previously (Constien et al, 2001), and applied the CDE diet protocol for 3 weeks (Fig. 1A,B) leading to toxic liver injury and intrahepatic cholestasis. Tamoxifen-mediated Cre activation in BECs is confirmed by tdTomato fluorescence resulting from the deletion of the floxed-Stop cassette in front of the tdTom locus. Successful homologous deletion of *Rage* in tdTomato-labeled *Hnf1b*-positive BECs is indicated by the onset of GFP expression from the Tk promoter (Fig. 1A). IF staining of biliary markers, CK19 and A6, showed that tdTomato is co-expressed selectively in CK19$^+$ and A6$^+$ BECs (Fig. EV1). Upon treatment with tamoxifen, *Rage* was specifically deleted in average in 60% of BECs in chronically injured mouse liver (Fig. 1C). By both IHC analyses of HNF1B and tdTomato expression and in situ hybridization of *Rage* on consecutively sectioned liver tissues, we confirmed that *Rage* expression is abolished only in BECs of *Hnf1b*-specific *Rage* knockout mice ($Rage^{\Delta BEC}$) (Fig. 1D).

The number of *Hnf1b*-positive BECs were substantially reduced upon *Rage* deletion in BECs when compared to $Rage^{WT}$ injured mice (Fig. 1D). The degree of DR was evaluated based on H&E staining, with the majority of $Rage^{WT}$ mice (~80%, total $n = 9$) and a moderate proportion of $Rage^{+/\Delta BEC}$ mice (40%, total $n = 10$) exhibited moderate to severe form of DR upon chronic injury, whereas DR in $Rage^{\Delta BEC}$ mice were substantially mitigated, with only 8.3% of the mice (total $n = 12$) showing a moderate level of DR (Fig. EV2F). Consistently, CK19 and A6 IF staining for BECs showed similar results (Fig. EV1).

The reduction of DR in $Rage^{\Delta BEC}$ mice may be due to decreased BEC proliferation. IF staining of the proliferation marker Ki67 suggested that RAGE controls BECs proliferation during injury, as Ki67 were expressed in BECs in CDE-challenged $Rage^{WT}$ and $Rage^{+/\Delta BEC}$, but not in $Rage^{\Delta BEC}$ mice (Fig. EV3). The biochemical serum marker of cholestasis, alkaline phosphatase (ALP), was elevated in $Rage^{WT}$ and to a lesser extent in $Rage^{+/\Delta BEC}$ and in $Rage^{\Delta BEC}$ mice upon CDE-induced injury (Fig. EV2E).

Thus, we hypothesize that RAGE regulates BEC expansion in a cell-autonomous manner to execute regulatory function in controlling the formation of DR in the context of liver injury (Pusterla et al, 2013).

### BEC-specific deletion of RAGE does neither interfere with inflammation nor steatosis

RAGE is a major regulator of the innate immunity and is crucial for sustaining inflammation in different disease models (Gąsiorowski et al, 2018; Gebhardt et al, 2008). Interestingly, in chronic liver diseases, the

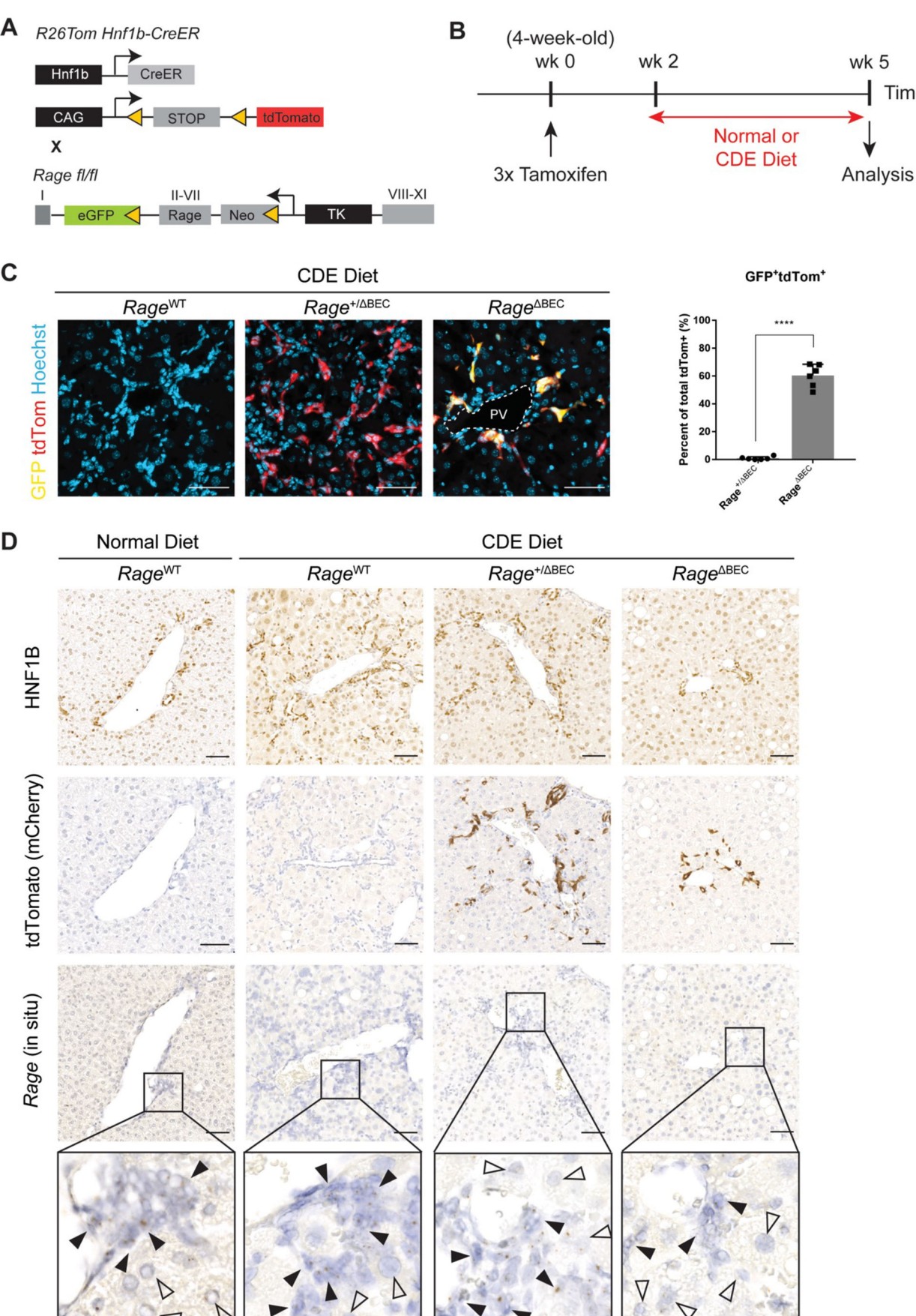

**Figure 1. Conditional deletion of *Rage* in BECs in a CDE diet-induced injury model of cholestasis.**

(A) Schematic diagram of generating a BEC-specific conditional *Rage* knockout mouse line. *R26^Tom^Hnf1b-CreER* reporter mouse was crossed with *Rage^fl/fl^* mouse that carries an eGFP reporter. (B) Schematic diagram of the experimental time line. A total of three doses of Tamoxifen was injected intraperitoneally every 3rd day into 4-week-old mice, which were subsequently kept for 2 weeks of wash-out period, followed by 3 weeks of CDE diet treatment. (C) Co-IF analysis of tdTomato and GFP staining to confirm genetic deletion of *Rage* in BEC in CDE-challenged mice, and quantification for tdTomato fluorescence and anti-GFP staining in CDE diet-treated *R26^Tom^Hnf1b-CreER Rage^+/fl^* (*Rage^+/ΔBEC^*) and *R26^Tom^Hnf1b-CreER Rage^fl/fl^* (*Rage^ΔBEC^*) mice. Data are shown as mean ± s.d. of n = 6 biological replicates per group. Two-tailed *t* test was used for statistical comparisons (****P < 0.0001). (D) Representative images of IHC staining of HNF1B, mCherry and in situ hybridization of *Rage* on consecutively sectioned liver tissues in normal diet-treated *Rage^WT^* mice or CDE diet-challenged *Rage^WT^*, *Rage^+/ΔBEC^* and *Rage^ΔBEC^* mice. ▲ indicates BECs; Δ indicates hepatocytes. Scale bar = 50 μm. Source data are available online for this figure.

proinflammatory transcriptomic profile is present in BECs in support of immune cell recruitment (Aguilar-Bravo et al, 2019); however, it remains unclear whether RAGE signaling in BECs is required for inflammation and immune response upon chronic injury.

To assess the impact of BEC-specific RAGE in CDE diet-induced inflammation in mouse livers, H&E staining and respective histological evaluation of normal and diseased livers was performed. *Rage^WT^*, *Rage^+/ΔBEC^* and *Rage^ΔBEC^* mice exhibited a rather comparable degree of lobular and portal inflammation in mouse livers regardless of the expression of RAGE in BECs (Fig. EV2A–C). Serum biochemical analysis of liver enzymes, alanine aminotransferase (ALT) and aspartate aminotransferase (AST), showed that RAGE in BECs did not affect the extent of liver damage as indicated by the abundance of CDE-induced serum marker levels (Fig. EV2D). IHC staining of Kupffer cells by Clec4f, macrophages by F4/80, and monocytes by CD68 showed that immune cell infiltration increased in CDE-induced injured mice regardless of the presence of RAGE in BECs (Fig. EV4). In line with previous results we obtained in the chronic biliary diseases Multidrug Resistance 2 (Mdr2) knockout model harboring a whole-body RAGE deletion (Pusterla et al, 2013), we confirmed that RAGE does not impact the recruitment of immune cells into injured liver.

We also investigated BEC- specific function of *Rage* in CDE-induced lipid accumulation by assessing the abundance of neutral triglycerides and lipids using Oil Red O staining, followed by histopathological analysis of hepatocellular steatosis. Using a semi-quantitative scoring system (Liang et al, 2014) we found that CDE diet-induced steatosis significantly; however, no significant difference of fat accumulation was observed comparing *Rage^WT^*, *Rage^+/ΔBEC^*, and *Rage^ΔBEC^* mice upon CDE-induced chronic injury (Fig. EV5).

In light of our previous data demonstrating RAGE to be a potential regulator of DR in the context of liver injury (Pusterla et al, 2013), we conclude that in contrast to its cell-autonomous function in BEC regulation, RAGE in BECs is not a prerequisite for orchestrating immune cell abundance during chronic liver injury.

## *Rage*-dependent signaling in BECs modulates extracellular matrix organization and triggers fibrosis

We sought to identify the *Rage*-dependent genetic program in BECs in response to chronic injury. Primary BECs were isolated from tamoxifen-treated, CDE-challenged mice, followed by FACS using a gating strategy based on high tdTomato (tdTom) and GFP expression for subsequent bulk RNA sequencing (Fig. 2A,B). Corresponding mice containing one intact RAGE allele (*Rage^+/ΔBEC^*), which express high levels of tdTom and only marginal levels of GFP, served as control

(*Rage* control) for the RNA-seq analysis. Differential gene expression analysis demonstrated that loss of *Rage* in BECs resulted in a significantly decreased expression of (i) extracellular matrix (ECM)-associated genes such as *Col4a1*, *Col16a1*, *Ltbp2* (latent transforming growth factor beta-binding protein 2), *Timp1* (tissue inhibitor of matrix metalloproteinase 1), *Cdh17* (Cadherin 17), and (ii) stem cell markers, such as *Cd44* and *Nes* (nestin) (Fig. 2C). KEGG and REACTOME pathway analyses showed that a majority of the enriched pathways (*P*-adjusted value < 0.05) of the differentially expressed genes were associated with ECM modulation and cell-cell interactions (Fig. 2D; Appendix Tables S3 and S4). Ingenuity pathway analysis (IPA) revealed that pathways, which are typically associated with hepatic fibrosis and hepatic stellate cell (HSC) activation were dependent on *Rage* in BECs (Fig. 2E). The expression of the differentially expressed genes in the enriched hepatic fibrosis/stellate cell activation and extracellular matrix organization pathways were visualized in heatmaps (Fig. 2F). Classical fibrotic mediators and markers (including *Tgfb1*, *Timp1*, and *Vcam1*), collagen of the ECM (including *Col4a1*, *Col4a2*, *Col16A1*, and *Col18a1*) and cell surface adhesion and signaling integrin (*Itga2*, *Itga5*) were found to be differentially expressed in the enriched hepatic fibrosis pathways (Appendix Fig. S2).

To validate the RNA-seq results related to fibrosis, we performed Picro-Sirius Red stain and found that CDE-challenged *Rage^WT^* and *Rage^+/ΔBEC^* mice displayed threefold more collagen fibers than *Rage^ΔBEC^* (Fig. 3A,B). Collateral histopathological evaluation also confirmed that the most advanced form of bridging fibrosis was observed in around 45% of *Rage^WT^* (n = 9) and 20% of *Rage^+/ΔBEC^* mice (n = 10), as compared to lower numbers in RAGE-deficient mice (~8% of n = 12). (Fig. 3C). Taken together, our histological analyses and RNA-seq data imply that the previously observed phenotype of compromised CDE-induced fibrosis in total RAGE knockout mice (Pusterla et al, 2013) is due to a critical trans-regulatory function of RAGE signaling in BECs.

Activation of portal fibroblasts and HSC is a well-established driving force of liver fibrosis, and the paracrine interplay between BECs and HSCs in the context of liver injury has been reported. To confirm our genome-wide transcriptomic data and histopathological analyses, we further investigate the function of RAGE in the crosstalk between BECs and HSCs by performing co-IF analysis of tdTom and markers of HSCs, Desmin and Vimentin. Under normal diet regime, only endothelial cells and pericytes surrounding the blood vessels were stained positive for both markers (Fig. 4A,B; Appendix Fig. S3). Upon CDE diet-induced injury, Desmin- and Vimentin-positive HSCs were markedly expanded into the liver parenchyma in *Rage^WT^* and *Rage^+/ΔBEC^* mice. Upon deletion of *Rage* in BECs, the number of Desmin- and Vimentin-

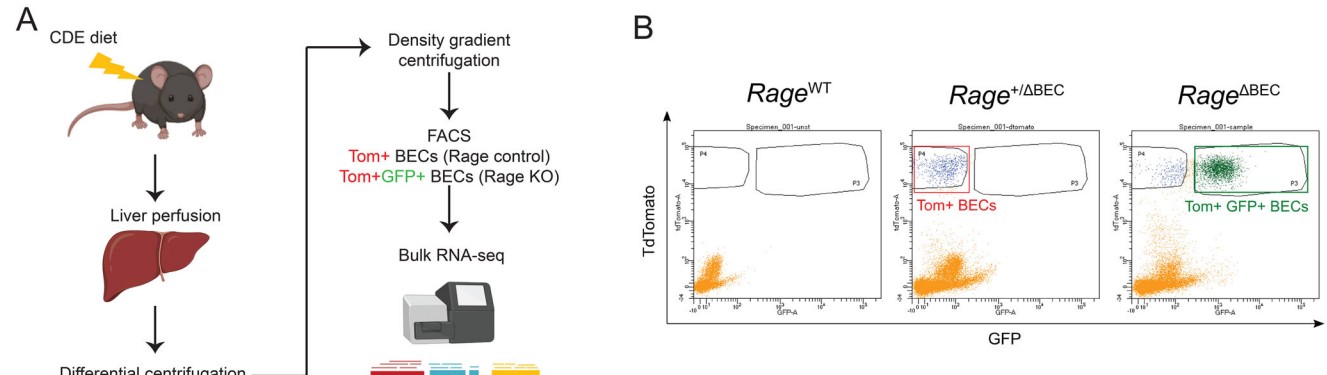

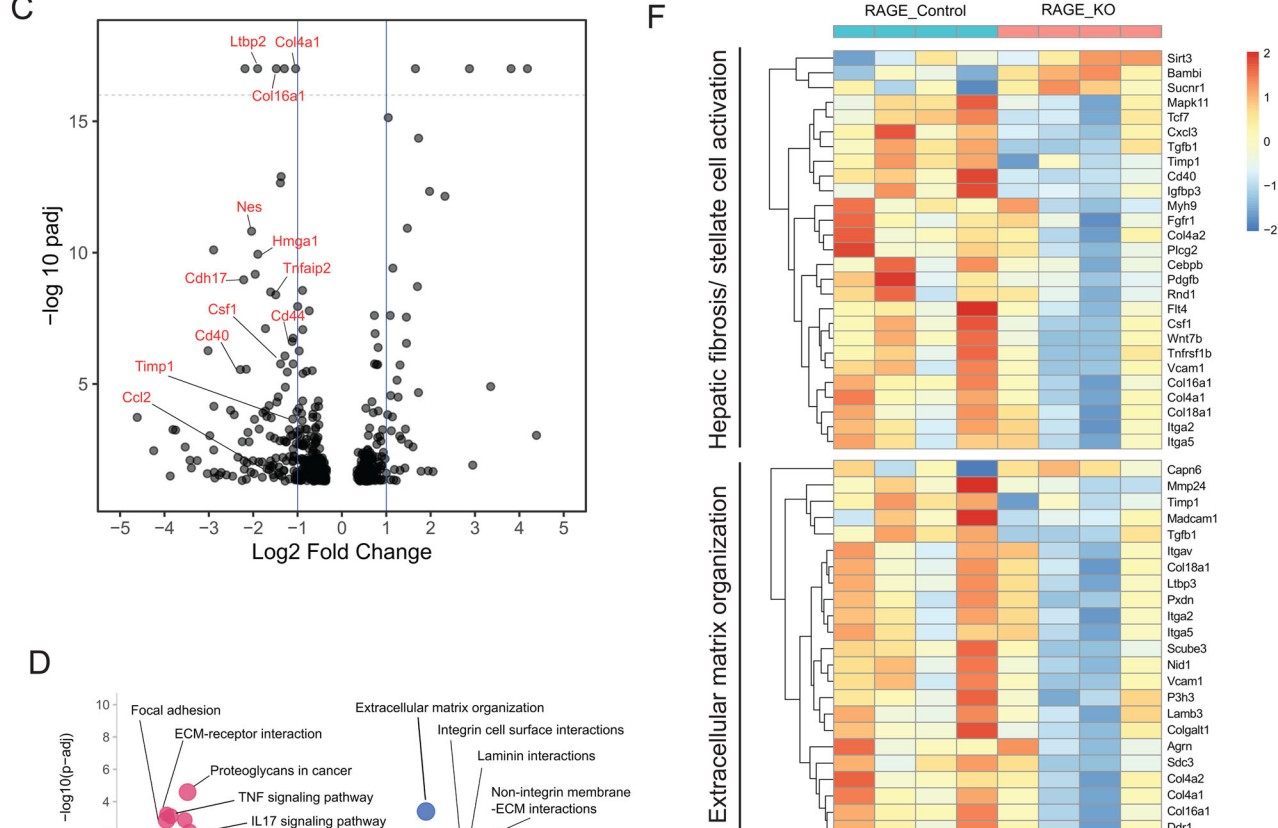

**Figure 2. RNA-seq data reveals the pro-fibrotic role of *Rage* in BECs during cholestasis.**

(A) Schematic diagram of the procedures to isolate primary BECs from chronically injured mice for bulk RNA-seq. (B) Representative FACS analysis of the primary BECs isolated from CDE diet-challenged *Rage*^WT^, *Rage*^+/ΔBEC^ and *Rage*^ΔBEC^ mice. tdTom+ BECs from *Rage*^+/ΔBEC^ mice ($n = 4$ independent biological replicates) and tdTom+GFP+ BECs from *Rage*^ΔBEC^ ($n = 4$ independent biological replicates) were sorted for direct RNA isolation followed by RNA-seq. tdTom+ BECs were taken as *Rage* control group for analysis. (C) Volcano plot of differentially expressed (DE) genes (*P*-adjusted value < 0.05) between *Rage* control and knockout BECs. The integrated DESeq2 tool (version 1.22.1) utilized normalized counts to calculate the log2 fold change and assess its statistical significance using the Wald test, reported as a *P* value. Adjusted *P* values (p-adj) were calculated using the Benjamini–Hochberg correction for multiple testing. Genes with a *P*-adj ≤0.05 were considered differentially expressed. (D) Enriched KEGG and REACTOME pathways (*P*-adjusted value < 0.05) of DE genes. Pathway analysis was performed with the R packages gprofiler2 (version 0.1.8) and AnnotationDbi (version 1.44.0). The DE gene list, together with their associated log2 fold channge were used as input for the pathway analysis. (E) Top 5 enriched canonical pathways of identified by Ingenuity Pathway Analysis (IPA) and the respective DE gene list. The DE gene list, together with their associated log2 fold change and *P*-adj were used as input for the IPA. (F) Heatmaps of the DE genes between *Rage* Control and knockout BECs in the corresponding pathways associated with hepatic fibrosis/stellate cell activation and extracellular matrix organization. Color scale bar represents regularized log transformed reads. Source data are available online for this figure.

positive cells was significantly reduced and accompanied by the reduction of tdTom+ BECs (Fig. 4A,B; Appendix Fig. S3). To further evaluate whether HSCs were indeed activated by RAGE activity in BECs in chronically injured mice, additional co-staining of both, BEC marker CK19 and αSMA highlighting activated HSCs was performed and showed consistent results (Fig. 4C). Only in *Rage*^WT^ and *Rage*^+/ΔBEC^ mice, but not in *Rage*^ΔBEC^ mice, activated αSMA-positive cells are clearly visible in the liver parenchyma in close proximity to BECs. Thus, we hypothesized that BECs may crosstalk with HSCs in a RAGE-dependent manner during liver injury and contribute to the activation of these cells.

## *Rage*-dependent soluble factors from BECs confer HSC activation

Intercellular communication is essential in orchestrating tissue homeostasis and functions. In direct cell-cell interaction, adjacent cells communicate through cell adhesion and diffusion of paracrine-soluble molecules. Indeed, our RNA-seq data of the primary BECs freshly isolated from CDE-challenged mice implied that *Rage* is regulating the focal adhesion and cell adhesion molecules in BECs (Fig. 2D), leading to subsequent signaling cascades in support of extracellular matrix remodeling. To investigate the mode of action of BEC-specific RAGE to affect BEC-HSC crosstalk in vitro, we isolated primary BECs from CDE diet-treated *Rage*^fl/fl^ mice to establish a *Rage* knockout cell line (Appendix Fig. S4). These cells were used for co-culturing with mCherry-labeled HSCs (MIM1-4HSC-mch). After four days of direct co-cultivation, cells were stained with eFluor660-conjugated antibodies against alpha-smooth muscle actin (α-SMA) representing an established marker of activated HSCs. Flow cytometry analyses (Fig. 5A–C) demonstrated that in the absence of BSCs already 50% of in vitro cultured HSCs exhibit a significant amount of α-SMA-positive cells (~50%). Nevertheless, when co-cultured with *Rage* wildtype (WT) BECs (referred as BEC-EV in Fig. 5) this further increased by ~25%. On the contrary, when HSCs were co-cultured with *Rage* knockout (KO) BECs (referred as BEC-Cre in Fig. 4), this increase of α-SMA expression in MIM1-4HSC is much smaller (10%) (Fig. 5C). These data are in strong support of a critical role of *Rage*-dependent signaling in BECs to activate HSCs either directly via direct cell contact or through *Rage*-dependent BEC-derived paracrine signals.

To experimentally address the presence of RAGE-dependent transacting paracrine factors from BECs acting on HSCs, conditioned medium from BECs were collected and placed onto HSCs. As a readout for HSC activation, measuring the status of vitamin A (retinol)-containing lipid droplets via BODIPY staining was used. In

quiescent HSCs, droplets are located within the cytoplasm. Upon activation, e.g., by treatment with recombinant TGFB1 used as a positive control, HSCs lose the lipid droplets and transdifferentiate into fibrogenic myofibroblasts, which is accompanied by a strongly reduced number of retinol-storing lipid droplets and an increased expression of α-SMA when compared to the untreated control (Figs. 6A and 5B). Using conditioned medium collected from RAGE WT BECs, a reduced number of retinol-storing lipid droplets and increased expression of α-SMA was also observed, although to a lower degree, as compared to TGFB1 treatment. In contrast, the HSCs remained inactivated when treated with conditioned medium collected from RAGE KO BECs. This phenotype was confirmed by measuring mRNA abundance of α-SMA (*Acta2*) and alpha-1 type I collagen protein (*Col1a1*) by qPCR. Again, we observed a higher expression of *Acta2* and *Col1a1* in HSCs treated with either rTGFB1 or conditioned medium from RAGE WT BECs, but not RAGE KO BECs (Fig. 6C).

Collectively, our results demonstrate the presence of a transregulatory RAGE-dependent signaling pathway in BECs that activate HSCs via paracrine signaling molecules. These molecules are likely to be part of the regulatory network of paracrine signaling between BECs and HSCs previously described to be essential in modulating the consequential fibrosis in liver disease models (Ishikawa et al, 2012; Wu et al, 2016).

## Deletion of *Rage* in BECs provides anti-fibrotic advantage by downregulating Notch activation

Next, we sought to identify the RAGE-dependent soluble factors from BECs that promote HSCs activation in *trans*. By analyzing the conditioned medium from *Rage* WT or KO BECs using mass spectrometry, we revealed that Notch ligand Jagged1 is differentially expressed in the secretome of the *Rage* WT versus KO BECs (Appendix Fig. S5). Interestingly, Jag/Notch signaling was described to play a key role in BEC-mediated liver regeneration during injury and aberrant activation of Notch signaling is evident in promoting fibrosis, hepatocellular carcinoma (HCC) and intrahepatic cholangiocarcinoma (ICC; (Geisler and Strazzabosco, 2015). Indeed, qPCR analyses show that *Jag1* mRNA expression was downregulated in *Rage*-deficient BECs (Fig. 7A). In agreement with these data, measuring protein levels by ELISA confirmed higher levels of Jagged1 in the conditioned medium of *Rage* WT BECs as compared to their KO counterparts (Fig. 7B). To validate the potential of Jagged1 as a promoter for Notch activation and HSC activation, HSCs were treated with recombinant Jagged1 in vitro. HSCs were activated by recombinant Jagged1 with

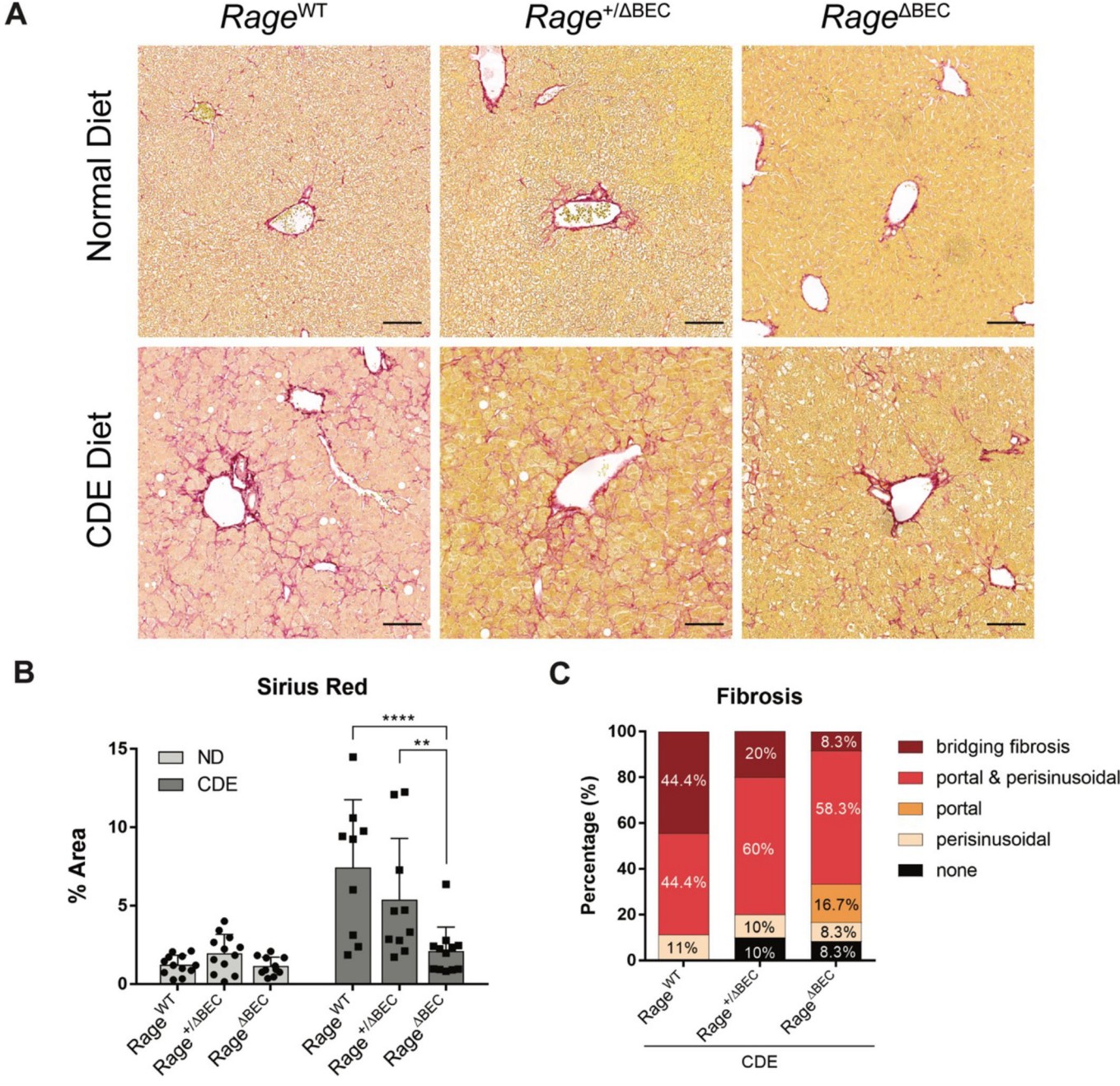

**Figure 3. *Rage* deficiency in BEC ameliorates fibrosis upon CDE-induced chronic injury.**

(A) Representative images showing histological Picro-Sirius Red staining (scale bar = 100 µm) and (B) corresponding quantification on liver sections from *Rage*$^{WT}$, *Rage*$^{+/\Delta BEC}$ and *Rage*$^{\Delta BEC}$ mice fed with normal diet (ND) or CDE diet. Data are shown as mean ± s.d. of biological replicates. For ND-treated *Rage*$^{WT}$ (n = 12), *Rage*$^{+/\Delta BEC}$ (n = 12), *Rage*$^{\Delta BEC}$ (n = 11); and CDE-treated *Rage*$^{WT}$ (n = 9), *Rage*$^{+/\Delta BEC}$ (n = 10), *Rage*$^{\Delta BEC}$ (n = 12). Two-way ANOVA with Turkey's multiple comparisons test was used for statistical comparison. (*P = 0.0232, ****P < 0.0001). (C) Histopathological evaluation of fibrosis in CDE-challenged mice based on Picro-Sirius Red staining. For CDE-treated *Rage*$^{WT}$ (n = 9), *Rage*$^{+/\Delta BEC}$ (n = 10), *Rage*$^{\Delta BEC}$ (n = 12). Source data are available online for this figure.

increased expression of αSMA and reduction of retinol-storing lipid droplets (Fig. 7C). Next, we utilized a complementary siRNA knockdown approach to silence *Jag1* using two independent siRNA sequences (siJag1 #1, #2) in *Rage* WT BECs. Upon silencing of *Jag1*, qPCR analysis showed an efficient knockdown of *Jag1* with a significant reduction of *Jag1* mRNA level (Fig. 8A). Further ELISA

analysis of secretory Jagged1 in the conditioned medium collected from BECs at 72 h after the siRNA knockdown validated the augmented release of secretory Jagged1 (Fig. 8B). Putting these conditioned media from BECs onto HSC reporter cells, silencing *Jag1* in BECs leads to a reduction of αSMA activation and a concomitant induction in retinol-storing lipid droplets in HSCs,

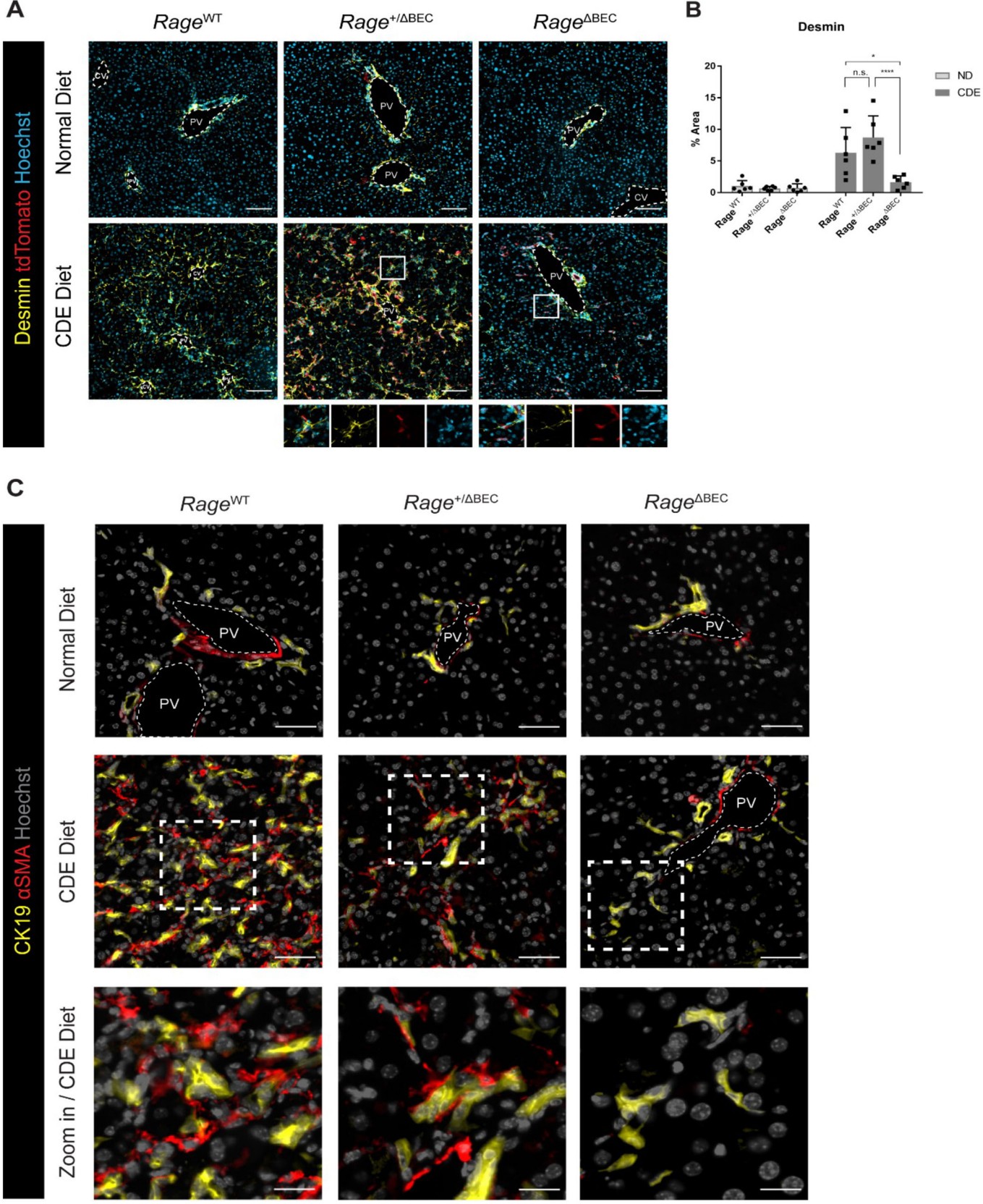

**Figure 4. BECs activate stellate cells in a *Rage*-dependent manner in vivo.**

(A) Representative images showing IF staining of portal fibroblast and stellate cell marker Desmin to assess the abundance of these cells in $Rage^{WT}$, $Rage^{+/\Delta BEC}$ and $Rage^{\Delta BEC}$ mice fed with normal or CDE diet. Scale bar = 100 µm. PV portal vein. (B) Quantification of percent area of Desmin staining (yellow) in (A). Data are shown as mean ± s.d. of $n = 6$ animals (biological replicates) per group. Two-way ANOVA with Turkey's multiple comparisons test was used for statistical comparison. (n.s. = not significant, *$P = 0.0161$, ****$P < 0.0001$). (C) Representative images showing co-IF staining of BEC marker, CK19, and activated portal fibroblast and stellate cell marker, αSMA, in $Rage^{WT}$, $Rage^{+/\Delta BEC}$ and $Rage^{\Delta BEC}$ mice fed with normal or CDE diet. $n = 2$ animals (biological replicates) were evaluated per group. Scale bar = 50 µm for images at the top and middle row. Scale bar = 20 µm for zoom in images at the bottom row. PV portal vein. Source data are available online for this figure.

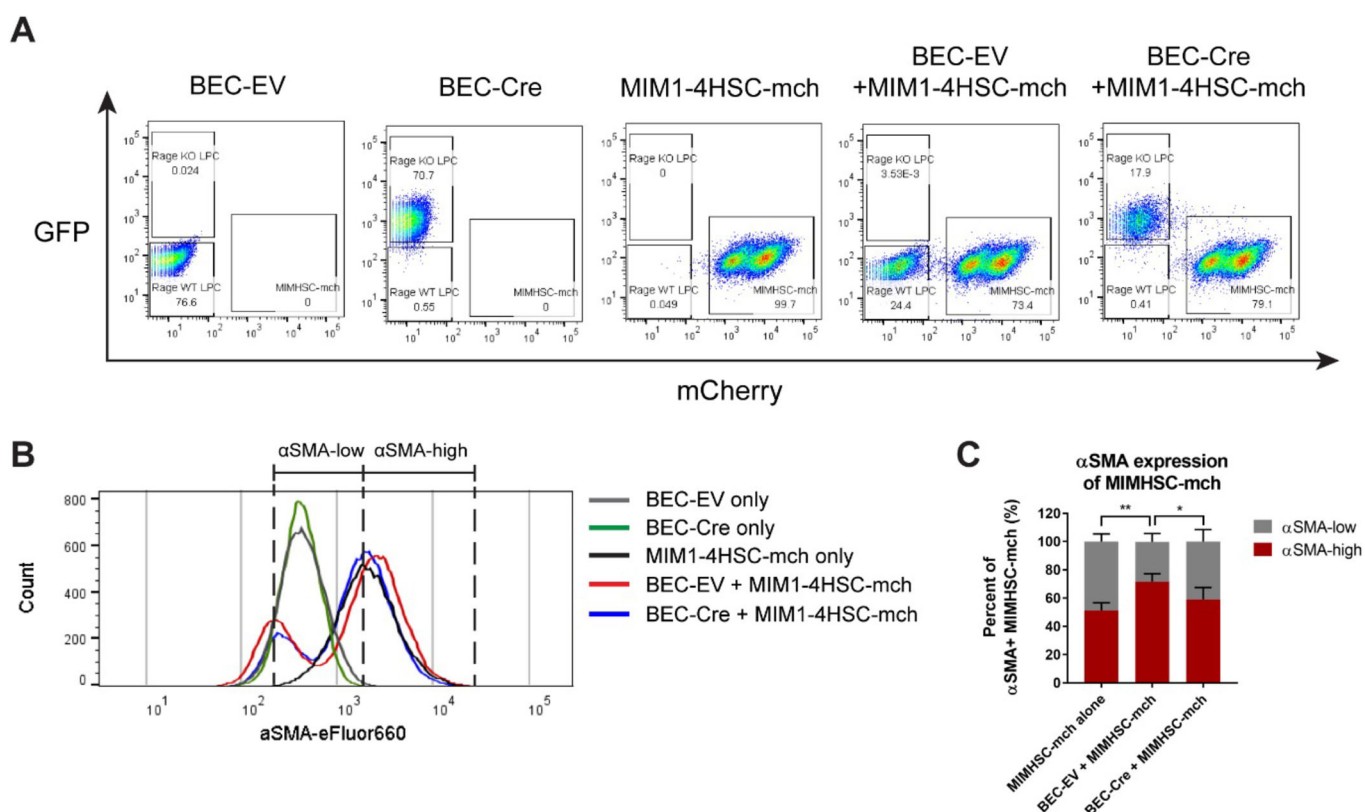

**Figure 5. BECs activate stellate cells in a *Rage*-dependent manner in vitro.**

Flow cytometry analysis of directly co-cultured BEC and MIM1-4HSC-mch stellate cells. (A) Representative flow cytometry gating strategy for BEC and MIM1-4HSC-mch that were directly co-cultured for 4 days. (B) Corresponding representative flow cytometry staggered histogram of α-SMA expression in BEC or MIM1-4HSC alone, or BECs that were co-cultured with MIM1-4HSC. (C) Percentage of α-SMA-low and α-SMA-high expression in MIM1-4HSC-mch when cultured alone, with *Rage* WT BECs or *Rage* KO BECs. Data are shown as mean ± s.d. of $n = 4$ independent experiments (biological replicates). Two-way ANOVA with Turkey's multiple comparisons test was used for statistical comparison. For α-SMA-high populations, **$P = 0.0011$, MIM1-4HSC-mch alone vs. *Rage* WT BECs+MIM1-4HSC-mch, *$P = 0.0402$, *Rage* WT BECs +MIM1-4HSC-mch vs. *Rage* KO BECs+MIM1-4HSC-mch; for α-SMA-low populations, **$P = 0.0011$, MIM1-4HSC-mch alone vs. *Rage* WT BECs+MIM1-4HSC-mch, *$P = 0.0377$, *Rage* WT BECs+MIM1-4HSC-mch vs. *Rage* KO BECs+MIM1-4HSC-mch. BEC-EV, BEC *Rage* WT; BEC-Cre, BEC *Rage* KO; MIM1-4HSC-mch, mCherry-labeled MIM1-4HSC. Source data are available online for this figure.

when compared to the treatment with conditioned medium from BECs harboring the siRNA negative control (Fig. 8C).

To confirm the relevance of the in vitro identified RAGE-dependent BEC-HSC interplay mediated by Jagged/Notch signaling in the in vivo setting, we performed staining of HES1 protein as a readout for Notch activation in the setting of our CDE diet injury model. IF staining of HES1, CK19 and Desmin revealed that HES1 expression is visible in CK19-positive BECs regardless of the genotype (Fig. 9). However, the number of Desmin-positive stellate cells expressing HES1 is reduced in mice harboring RAGE-deficient BECs, as compared to $Rage^{WT}$ or $Rage^{+/\Delta BEC}$ mice (Fig. 9).

Collectively, our data imply that RAGE in BECs is required for producing Jagged1 to establish intercellular communication between BECs and HSCs thereby potentiating and maintaining Notch-mediated HSC activation in chronic liver injury.

## Discussion

In this study, we employed the well-established CDE diet-induced injury model to demonstrate the direct pathophysiological role of RAGE in DR-associated liver fibrosis. This protocol is not a

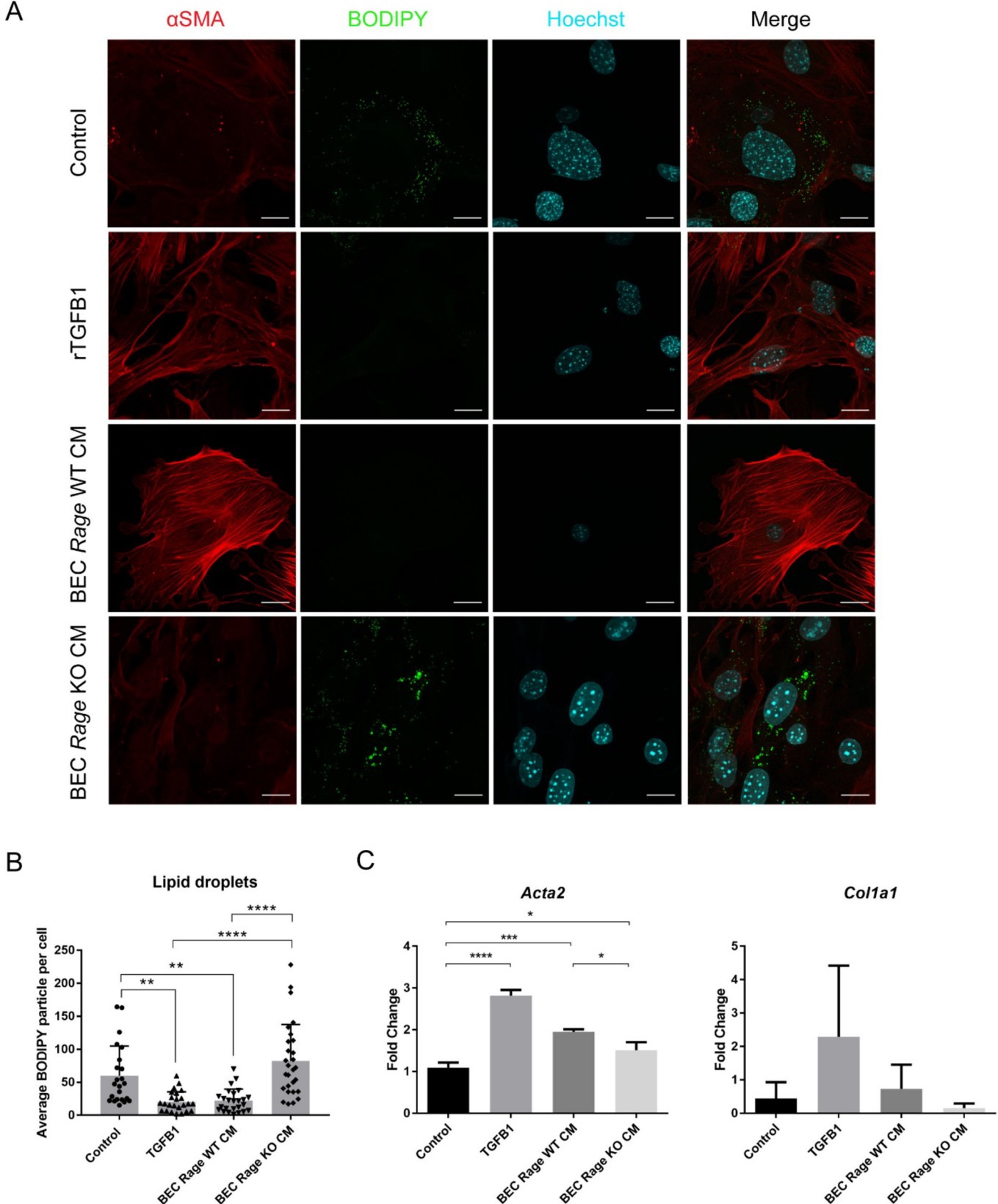

◀

**Figure 6. BECs induce HSC activation by *Rage*-dependent paracrine signals.**

(A) IF staining of α-SMA for myofibroblastic actin filaments and BODIPY for retinol (vitamin A)-containing lipid droplets in MIM1-4HSC that were treated with 5 ng/ml recombinant TGFB1, or conditioned medium (CM) collected from *Rage* WT or *Rage* KO BECs for 48 h. $n = 3$ independent experiments were performed. Scale bar = 20 µm. (B) Quantification of the number of BODIPY$^+$ particle per cell in treated MIM1-4HSC. $n = 5$ independent experiments were performed. Data are shown as mean ± s.d. of five technical replicates per $n = 5$ independent experiments collected. One-way ANOVA with Dunnett's multiple comparisons test was used for statistical comparison. **$P = 0.0032$, Control vs. TGFB1; **$P = 0.0052$, Control vs. BEC *Rage* WT CM; ****$P < 0.0001$, TGFB1 vs. BEC *Rage* KO CM, BEC *Rage* WT CM vs. BEC *Rage* KO CM. (C) mRNA expression of *Acta2* and *Col1a1* in treated MIM1-4HSC. Data are shown as mean ± s.d. of $n = 3$ independent experiments (biological replicates). One-way ANOVA with Turkey's multiple comparisons test was used for statistical comparison. ****$P < 0.0001$, Control vs. TGFB1; ***$P = 0.0003$, Control vs. BEC *Rage* WT CM; *$P = 0.0225$, Control vs. BEC *Rage* KO CM; *$P = 0.0179$, BEC *Rage* WT CM vs BEC *Rage* KO CM. Source data are available online for this figure.

classical "cholestasis model", as extrahepatic and intrahepatic bile ducts are primarily not affected but intrahepatic cholestasis is present due do hepatocyte damage with disruption of the canalicular architecture/connectivity/function to Canals of Hering. Previous studies proposed that RAGE is required for DR in both autophagy-deficient mice and spontaneous biliary fibrosis model (Khambu et al, 2018; Pusterla et al, 2013), but did neither conclusively clarify the identity of the cell type mediating RAGE-dependent BEC activation and expansion, nor the molecular mechanisms of RAGE-mediated control of fibrosis. In this study, we found that RAGE in BECs is required for the activation and expansion of these cells into the parenchyma upon chronic liver injury in a cell-intrinsic manner. This study further highlighted the significance of BEC-specific RAGE for activation of HSCs and portal fibroblasts and ECM deposition to promote liver fibrosis.

HSCs are central mediators of chronic liver injury and hepatocarcinogenesis. The balance between subpopulations of HSCs, such as the quiescent cytokine-producing HSC and activated myofibroblastic HSC, during chronic liver diseases is associated with the status of HCC development (Filliol et al, 2022). In the perspective of the biliary tree-associated disease states, the interplay between activated HSCs and BECs plays a dynamic role in DR-mediated liver regeneration and fibrosis. For instance, activated HSC-derived paracrine factors can evoke a protective response by inducing BEC-mediated liver regeneration (Chang et al, 2017; Pintilie et al, 2010; Kordes et al, 2014). In a classical 2-acetylaminofluorene/partial hepatectomy (2AAF/PH)-induced BEC response model, the inhibition of HSC activation by administration of a L-cysteine diet suppresses ductular response during the process of liver regeneration, suggesting that HSCs are required for BEC expansion (Pintilie et al, 2010). A more recent study also showed that HSC-derived growth factors trigger BEC regenerative response and ameliorates liver fibrosis (Dai et al, 2019). On the contrary, some studies have also proposed that BEC proliferation exacerbates fibrosis (Kuramitsu et al, 2013; Chobert et al, 2012). Nonetheless, to date, the direct influence of BECs on HSCs has not been reported. In line with previous reports (Paku et al, 2001; Pintilie et al, 2010; Van Hul et al, 2009), we observed BECs and HSCs to be localized in close proximity with each other upon CDE-induced injury, suggesting a crosstalk between these two liver resident cell types.

Our results have provided novel insights into the impact of ductular response on HSC activation in supporting fibrosis. Although, activated HSCs are the dominant contributor of ECM production that causes fibrosis in mouse models of toxic, cholestatic and fatty liver diseases (Mederacke et al, 2013), our

RNA-seq data showed ECM organization genes were upregulated in primary BECs isolated from Rage control mice when compared to the mutant counterpart (Fig. 2F). Interestingly, mesenchymal cell markers such as CD44 and Thy-1 were downregulated in BECs from *Rage*$^{ΔBEC}$ mice (Appendix Fig. S6), suggesting that in the presence of RAGE BECs exhibit a more mesenchymal-like status and may also contribute to production of ECM in a RAGE-dependent manner. Furthermore, our RNA-seq data and in vivo analyses suggest that RAGE-mediated ductular response preceded HSC activation and fibrosis. Of note, we observed a remarkable correlation in the degree of DR and HSCs activation, demonstrating that RAGE-driven DR directly affect the cellular and molecular role of HSCs. Thus, following the report by Filliol and colleagues (Filliol et al, 2022), it will be interesting to define on the molecular level whether RAGE-controlled DR causes a similar shift in the HSC subpopulations towards the myofibroblastic phenotype, thereby exacerbating fibrosis in early chronic liver diseases state.

Notch signaling is an evolutionarily conserved pathway that is essential for cell fate decision, differentiation and homeostasis. During liver development, notch signaling is particularly crucial for the biliary lineage specification and biliary tree development. Under pathological conditions, aberrant activation of Notch signaling is frequently implicated in hepatocellular carcinoma (Villanueva et al, 2012; Zhu et al, 2021) and intrahepatic cholangiocarcinoma (Zender et al, 2013). Mutations in notch ligand *JAG1* or receptor *NOTCH2* harbors a rare autosomal dominant genetic disorder, namely Alagille syndrome (ALGS), which is characterized by intrahepatic bile duct paucity in infants and children, leading to jaundice and cholestasis. In this study, we showed that Rage deficiency in BECs significantly reduces the level of secretory Jagged1 in vitro and hampered DR in vivo, which resembles the development of ALS due to JAG1 mutation. Given the central role of Jagged1 in ALGS, our findings suggested that RAGE may be a modulator of Jagged1 that contributes to the development of biliary disease.

Notch signaling in ductular reaction and fibrosis have been extensively described in experimental mouse injury models. For instance, Notch signal is important for determining the biliary lineage cell fate and expansion in an acute hepatocyte senescence *Mdm2* deletion model (Minnis-Lyons et al, 2021) and CDE and DDC-induced cholestatic models (Boulter et al, 2012). Aberrant Jagged1 activity from hepatocytes is likely to be mediated by TLR4-NFκB signaling and is necessary for Notch activation in NASH-induced liver fibrosis (Yu et al, 2021). To resolve Notch-induced hepatic injury, inhibition of Notch components seemingly has a protective effect on hepatocytes and ameliorates HSCs activation and fibrosis as described in two experimental models of carbon

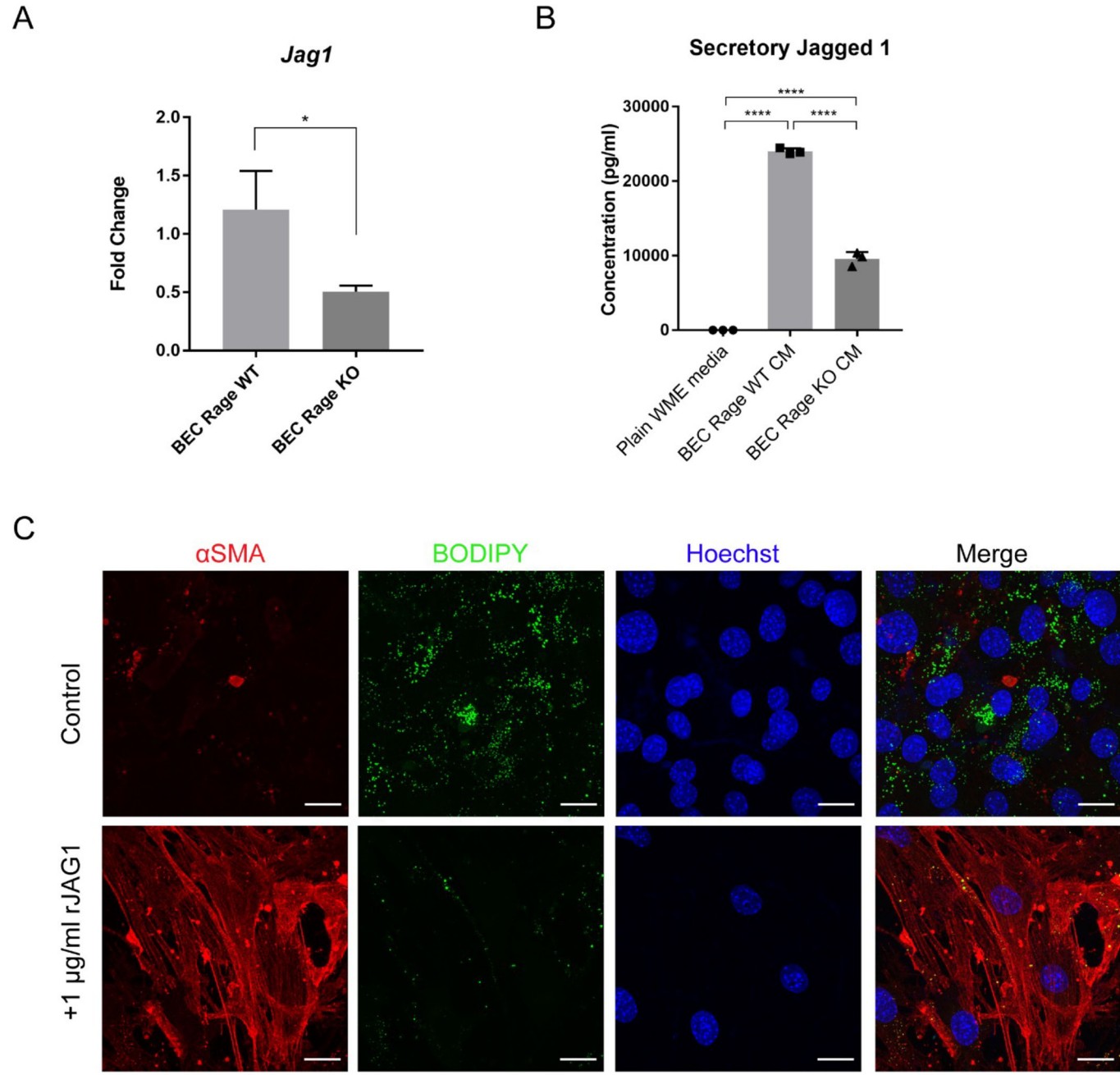

**Figure 7. BEC-derived secretory JAG1 activates Notch signaling in HSCs.**

(A) Endogenous mRNA expression of *Jag1* in *Rage* WT and KO BECs. Data are shown as mean ± s.d. of *n* = 3 independent biological replicates. Two-tailed *t* test was used for statistical comparisons (*P = 0.0222). (B) ELISA measurement of Jagged1 in BEC basal culture medium (Control) and conditioned medium (CM) collected from *Rage* WT and KO BECs. Data are shown as mean ± s.d. of *n* = 3 independent biological replicates. One-way ANOVA with Turkey's multiple comparison test was used for statistical comparisons (****P < 0.0001). (C) Representative images of IF staining of αSMA and BODIPY in HSCs treated with PBS control or 1 μg/ml of recombinant Jagged1 (rJAG1). *n* = 3 independent experiments were performed. Scale bar = 20 μm. Source data are available online for this figure.

tetrachloride hepatotoxin-induced liver injury (Bansal et al, 2015; Chen et al, 2012). Previous work has opposing results about the source of Jagged1 in injury models, with one study suggested that Jagged1 is produced by BEC-associated myofibroblast (Boulter et al, 2012), while another study suggested that it is produced by BECs themselves (Minnis-Lyons et al, 2021). In addition, the nature of the upstream regulator contributing to Jagged/Notch signaling is presently unknown, and whether or not the expansion of BECs is required for supporting HSCs differentiation have not been investigated. In this study, we revealed that BEC-derived secretory

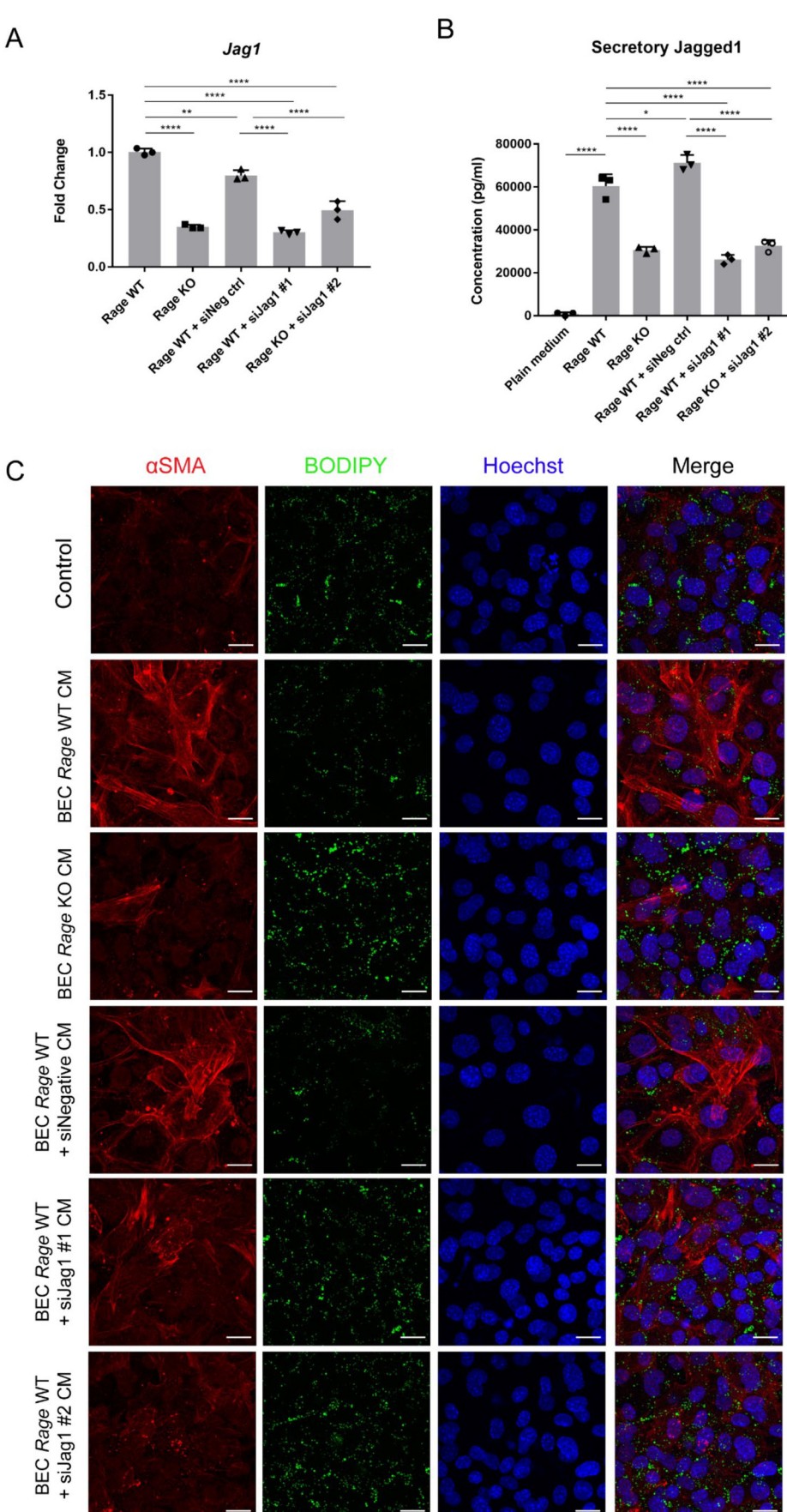

**Figure 8. RNAi knockdown of *Jag1* in BECs augments HSC activation in trans.**

(A) qPCR analysis of mRNA expression of *Jag1* in *Rage* WT or KO BECs, or *Rage* WT BECs treated with siRNA negative control (siNeg Ctrl), or siJag1 clone 1 (siJag1 #1) or siJag1 clone 2 (siJag1 #2) for 72 h. Data are shown as mean ± s.d. of $n = 3$ independent biological replicates. One-way ANOVA with Turkey's multiple comparison test was used for statistical comparisons. **$P = 0.0015$, ****$P < 0.0001$. (B) ELISA measurement of secretory Jagged1 in *Rage* WT or KO BECs, or *Rage* WT BECs treated with siRNA negative control (siNeg Ctrl), or siJag1 clone 1 (siJag1 #1) or siJag1 clone 2 (siJag1 #2) for 72 h. Data are shown as mean ± s.d. of $n = 3$ independent biological replicates. One-way ANOVA with Turkey's multiple comparison test was used for statistical comparisons (*$P = 0.0106$, ****$P < 0.0001$). (C) IF staining of αSMA and BODIPY in HSCs treated with plain William's E medium (control), conditioned medium (CM) collected from BEC Rage WT or KO cells, or CM collected from BEC Rage WT treated with siNeg, siJag1 #1 or siJag1 #2 for 72 h. Scale bar = 20 μm. Source data are available online for this figure.

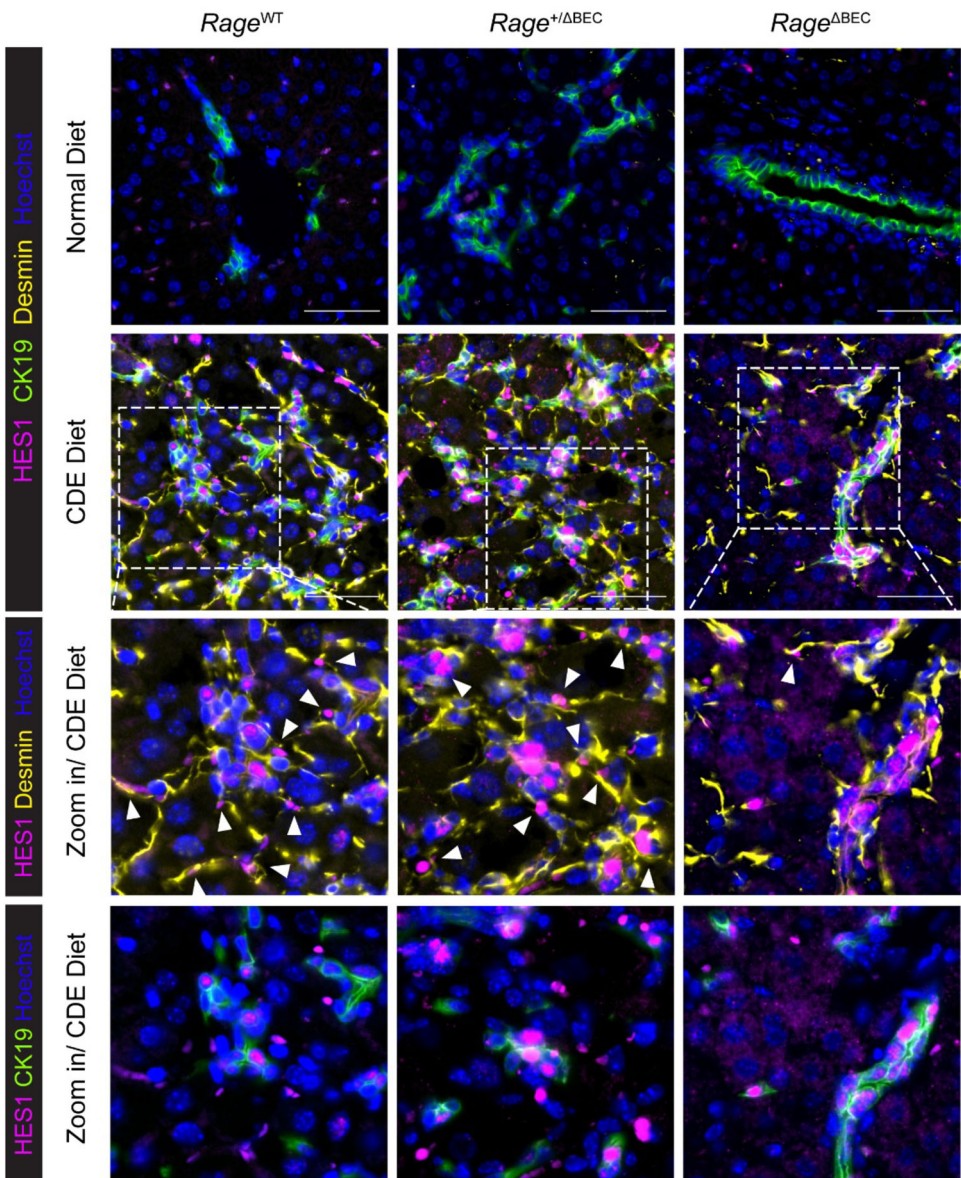

**Figure 9. BEC-specific RAGE activates Notch signaling in BECs and potentiates HSC activation in cholestatic mice.**

Multiplex IF staining of HES1, CK19 and Desmin in *Rage*^WT, *Rage*^+/ΔBEC and *Rage*^ΔBEC mice fed with normal or CDE diet. HES1 expressed by HSCs is indicated by white arrow (third row). $n = 3$ independent biological replicates were performed per group. Scale bar = 50 μm. Source data are available online for this figure.

Jagged1 ligand is controlled by RAGE activity. The expression of *Jag1* in the wildtype BECs are indeed higher than the *Rage* KO BECs. On the one hand, the paracrine secretory ligands from BECs are sufficient to induce the transdifferentiation of HSCs into myofibroblast and triggered Notch activation in HSCs in vitro. On the other hand, the very close proximity of BECs and a-SMA-positive HSCs in CDE-treated mice (Fig. 4C) implies that also membrane-bound Jagged1 on BECs is capable of binding to its

ligands on HSCs. Regardless of the exact mode of action, we speculated that BECs carry a pro-fibrotic role in supporting HSCs activation via Notch signaling during liver injury. Indeed, we observed a robust ductular response concomitantly with a more aggressive fibrotic phenotype in our animal model. Biliary-specific *Rage* deletion ameliorated diet-induced fibrosis with reduced Notch activity as demonstrated by HES1 staining. In line with previous studies, Notch signaling was upregulated in activated HSCs and *Hes1* is transcriptionally active during fibrosis progression (Bansal et al, 2015; Yang et al, 2019). Taken together, BEC-induced Notch activation in HSCs, possibly in conjunction with other previously described cholangiokines including TGFß (Cai et al, 2023) may represent a major causal effect of fibrosis during cholestasis, and such crosstalk is dependent on biliary-specific RAGE activity.

Chronic intrahepatic cholestatic diseases commonly result from impaired bile secretion or biliary phospholipid secretion (Srivastava, 2014). Patients exhibiting this phenotype are at high risk of developing periportal fibrosis. Progression to biliary cirrhosis with prominent proliferating bile ducts and liver failure are commonly observed at advanced stage of the disease, which ultimately leads to hepatocellular carcinoma or cholangiocarcinoma (Ibrahim et al, 2022; Srivastava, 2014). Recently in 2021, two ileal bile acid transporter (IBAT) inhibitors, maralixibat and odevixibat, have been approved for the treatment of pruritus in patients with Alagille Syndrome or Progressive Familial Intrahepatic Cholestasis (PFIC), respectively, and showed efficacy in reducing the intrahepatic retention of bile acids (Miethke et al, 2024; Ovchinsky et al, 2024). Latest studies on the usage of FXR agonists and FGF19 analogs to reduce bile acid synthesis have also shed light on anti-cholestatic management (Stedman et al, 2006; Harrison et al, 2022). Nevertheless, the additional benefits of these potential agents on complications arise from cholestasis would require further investigations and monitoring. To date, there are still a limited FDA-approved therapeutic options to treat or prevent cholestasis-induced chronic diseases, by so far only liver transplantation is served as a complete resolution to abrogate the risk of liver failure and tumorigenic progression (Ibrahim et al, 2022). In the present study, we showed that RAGE is expressed mainly on BECs and functions as a master regulator of cholestasis-induced DR and fibrosis. In view of the cell type-specific function of RAGE in the context of cholestasis, RAGE may serve as a therapeutic target for preventing cholestasis and/or associated fibrosis. To date, there are only a few known RAGE small molecular inhibitors available for preclinical and early clinical trials in various inflammatory diseases. For instance, Azeliragon, the orally bioavailable small molecule inhibitor of RAGE, was used as an agent in preclinical studies of diabetic complications (Ma et al, 2023) or clinical studies of Alzheimer's disease (Galasko et al, 2014; Burstein et al, 2014) and showed efficacy. FPS-ZM1, a small molecule that is able to inhibit the interaction between RAGE and its ligands, was shown to reduce inflammation in experimental mouse models (Hudson and Lippman, 2018). In view of its efficacy in preclinical studies of other inflammatory diseases, the RAGE inhibitors Azeliragon or FPS-ZM1 might have the potential to be used as pharmacological tools in treating cholestasis or cholestasis-associated fibrosis. However, the mode of action of the RAGE inhibitors in cholestatic diseases are yet to be explored and elucidated.

# Methods

## Reagents and tools table

| Reagent/resource | Reference or source | Identifier or catalog number |
|---|---|---|
| **Experimental models** | | |
| C57BL/6 (*M. musculus*) | Charles River | |
| *Rage* +/fl (*M. musculus*) | Pusterla et al, 2013 | |
| *Rage* fl/fl (*M. musculus*) | Constien et al, 2001 | |
| R26^Tom Hnf1b-CreER (*M. musculus*) | Jörs et al, 2015 | |
| R26^Tom Hnf1b-CreER Rage^+/fl (*M. musculus*) | This study | |
| R26^Tom Hnf1b-CreER Rage^fl/fl (*M. musculus*) | This study | |
| HEK293T | American Type Culture Collection (ATCC) | |
| BEC RAGE WT | This study | |
| BEC RAGE KO | This study | |
| MIM1-4HSC | Proell et al, 2005 | |
| MIM1-4HSC-mCherry | This study | |
| **Recombinant DNA** | | |
| pUltra-Hot (3^rd generation) | Addgene | 24130 |
| pMXpie-EV | This study | |
| pMXpie-Cre | This study | |
| **Antibodies** | | |
| A6 | Developmental Studies Hybridoma Bank | AB_2618041 |
| BODIPY™ 493/503 | Invitrogen | D3922 |
| CK19 | Abcam | ab133496 |
| Clec4f | R&D Systems | AF2784-SP |
| Desmin | Abcam | ab15200 |
| F4/80 | Linaris | T-2006 |
| GFP | Abcam | ab13970 |
| HES1 | MBL International Corporation | D134-3 |
| HNF1B | Proteintech | 12533-1-AP |
| Hoechst 33342 | Biomol | ABD-17530 |
| Ki67 | Abcam | ab15580 |
| mCherry | Abcam | ab167453 |
| Vimentin | Abcam | ab92547 |
| αSMA | Abcam | ab5694 |
| αSMA-eFluor660 | eBioscience | 50-9760-82 |
| LIVE/DEAD™ Fixable Aqua Dead cell stain | Invitrogen | L34957 |
| Biotinylated Goat α-rabbit IgG | Vector Laboratories | BA-1000 |
| Biotinylated Goat α-rat IgG | Vector Laboratories | BA-9400 |
| Goat α-rabbit Alexa 647 | Invitrogen | A21244 |
| Goat α-rat Alexa 647 | Invitrogen | A21247 |
| Goat α-chicken Alexa 647 | Invitrogen | A21449 |

| Reagent/resource | Reference or source | Identifier or catalog number |
|---|---|---|
| FlexAble CoraLite Plus 647 Antibody labeling kit for Rabbit IgG | Proteintech | KFA003 |
| FlexAble CoraLite Plus 750 Antibody labeling kit for Rabbit IgG | Proteintech | KFA004 |
| Streptavidin, CY3 | Vector Laboratories | SA-1300-1 |
| **Oligonucleotides and other sequence-based reagents** | | |
| PCR primers for genotyping | This study | Appendix Table 1 |
| Mouse primers used for qPCR analysis | This study | Appendix Table 2 |
| Silencer™ Select Negative Control No. 1 siRNA | Invitrogen | 4390843 |
| Anti-Jagged1 siRNA#1 | Invitrogen | ID s68530 |
| Anti-Jagged1 siRNA#2 | Invitrogen | ID s68532 |
| RNAscope probe #Mm-Rage-O3 | Advanced Cell Diagnostics | 882411 |
| **Chemicals, enzymes, and other reagents** | | |
| Bovine serum albumin (BSA) | Sigma-Aldrich | A9418 |
| Calcium chloride ($CaCl_2$) | Merck | 102378 |
| Choline-deficient diet | Research Diet Inc. | A16030901 |
| Collagenase Type IV | Sigma-Aldrich | C5138-1G |
| DL-Ethionine | Sigma-Aldrich | E5139 |
| Dulbecco's Modified Eagle's Medium (DMEM) – high glucose | Sigma-Aldrich | D5671 |
| Dulbecco's Phosphate Buffered Saline (DPBS) | Pan Biotech | P04-36500 |
| Eosin 1% | Morphisto | 10177 |
| Epidermal Growth Factor, Mouse Natural (Culture Grade) (EGF) | Corning | 354001 |
| Ethylene glycol tetraacetic acid (EGTA) | Sigma-Aldrich | 324626 |
| Fetal Bovine Serum (FBS) | Sigma-Aldrich | F4135 |
| Gibco™ L-Glutamine | Thermo Fisher Scientific | 25030081 |
| Gibco™ Williams' Medium E | Thermo Fisher Scientific | 22551089 |
| Glucose | Sigma-Aldrich | G7528 |
| Hank's balanced salt solution (HBSS) | Thermo Fisher Scientific | 14065056 |
| Hematoxylin acidic after MAYER | Morphisto | 16133 |
| Heparin Sodium Salt | Sigma-Aldrich | H3393-50KU |
| HEPES | Sigma-Aldrich | H3375-500G |
| Insulin, Human | Sigma-Aldrich | I2643 |
| Insulin-like Growth Factor-II, Human (IGF2) | Sigma-Aldrich | I2526 |
| KETASET | Zoetis | 043-304 |
| Magnesium sulfate ($MgSO_4$) | Sigma-Aldrich | M7506 |
| Oil Red O | Thermo Fisher Scientific | A12989 |

| Reagent/resource | Reference or source | Identifier or catalog number |
|---|---|---|
| OptiMEM Reduced Serum Medium | Thermo Fisher Scientific | 31985062 |
| Penicillin/Streptomycin Solution | Sigma-Aldrich | P4333 |
| Percoll® | Sigma-Aldrich | P1644 |
| Picro-Sirius Red | Morphisto | 13422 |
| Polyethylenimine (PEI) | Sigma-Aldrich | 408727 |
| Potassium chloride (KCl) | Carl Roth | 6781.1 |
| Potassium dihydrogen phosphate ($KH_2PO_4$) | Carl Roth | 3904.1 |
| *Power* SYBR™ Green PCR-Master Mix | Thermo Fisher Scientific | 4367659 |
| Pronase | Roche | 10165921001 |
| Recombinant human TGF-beta 1 Protein | Bio-Techne | 7754-BH |
| Recombinant rat Jagged1 Fc chimera protein, CF | Bio-Techne | 599-JG-100 |
| Rompun® 2% | Bayer Animal Health GmbH | 17033-099-05 |
| ROTI® Histofix 4% formaldehyde | Carl Roth | P087.1 |
| Sodium chloride (NaCl) | Carl Roth | 3957.1 |
| Sodium dihydrogen phosphate ($NaH_2PO_4$) | Carl Roth | 1H52.4 |
| Sodium hydrogen carbonate ($NaHCO_3$) | Merck | 106329 |
| Sucrose | Sigma-Aldrich | S7903 |
| Sunflower Seed oil | Sigma-Aldrich | S5007 |
| Tamoxifen | Sigma-Aldrich | T5648 |
| Tissue-Tek O.C.T. Compound | Sakura Finetek USA | 4583 |
| Triton X-100 | Sigma-Aldrich | X-100 |
| **Software** | | |
| BD FACSDiva™ Software | Becton Dickinson Biosciences | |
| FlowJo v10 | Tree Star | |
| GraphPad Prism 7.03 | GraphPad Software | |
| Image J | https://imagej.net/ij/index.html | |
| Primer-BLAST | https://www.ncbi.nlm.nih.gov/tools/primer-blast/ | |
| StepOne Software v2.3 | Thermo Fischer Scientific | |
| ZEN 2.3 (Blue edition) | Zeiss | |
| **Other** | | |
| BD Cytofix/Cytoperm™ Fixation/Permeabilization Solution Kit | BD Biosciences | AB_2869008 |
| DAB Peroxidase Substrate kit | Vector Laboratories | SK-4100 |
| FuGENE HD transfection reagent | Promega | E2311 |

| Reagent/resource | Reference or source | Identifier or catalog number |
|---|---|---|
| Lipofectamine® RNAiMAX reagent | Thermo Fisher Scientific | 13778100 |
| Rat Jagged1 ELISA Kit | Thermo Fisher Scientific | ERA35RB |
| RNAscope™ 2.5 HD Brown Kit | Advanced Cell Diagnostics | 322371 |
| RNase-Free DNase Set | Qiagen | 79254 |
| RNeasy Mini Kit | Qiagen | 74104 |
| TSA Biotin System | Perkin Elmer | NEL700A001KT |
| VECTASTAIN® Elite® ABC HRP Kit | Vector Laboratories | PK-6100 |
| Axio Scan.Z1 Slide Scanner | Zeiss | |
| CLARIOstar microplate reader | BMG Labtech | |
| FACSAria cell sorter | BD Biosciences | |
| HiSeq 4000 sequencing system | Illumina | |
| LSM 710 Laser Scanning Confocal Microscope | Zeiss | |
| LSRFortessa™ cell analyser | BD Biosciences | |
| Orbitrap Exploris 480 mass spectrometer | Thermo Fisher Scientific | |
| StepOnePlus™ Real-Time PCR System | Applied Biosystems | |

## Animal study

The tamoxifen-inducible $R26^{Tom}$ $Hnf1b$-$CreER$ mouse strain and mice that carry floxed alleles for $Rage$ ($Rage^{fl/fl}$) have been described previously (Jörs et al, 2015). $R26^{Tom}$ $Hnf1b$-$CreER$ $Rage^{+/fl}$ was used to obtain the BEC-specific $Rage$ knockout mouse strain $R26^{Tom}Hnf1b$-$CreER$ $Rage^{fl/fl}$ that expresses the fluorescent protein tdTomato and GFP upon Cre recombination. $R26^{Tom}Rage^{+/+}$ was used for Rage WT. $R26^{Tom}$ $Hnf1b$-$CreER$ $Rage^{+/+}$ gave very similar results. Genotyping was performed after the mice were weaned and before tamoxifen injection. Sequences of primers used for genotyping are available in Appendix Table S1. To induce Cre activation in transgenic mice, tamoxifen (Sigma) was dissolved in a sunflower seed oil/ethanol (9:1) mixture to a final concentration of 20 mg/ml, and administered to 4- to 5-week-old mice at 100 mg/kg body weight every third day for three doses in total by intraperitoneal injection. The mice were maintained for further 2 weeks of wash-out period prior to the start of injury model. To induce chronic liver injury, mice were fed with choline-deficient, ethionine-supplemented (CDE) diet (Research Diet) for 3 weeks to induce chronic liver injury (Pusterla et al, 2013). Mice were maintained on a 12-h dark/12-h light cycle with ad libitum access to food and water. All animal experiments were approved by the responsible authority for animal experiments (Regierungspräsidium Karlsruhe, Germany) and performed in conformity with the German Law for Animal Protection.

## Liver perfusion

Livers were perfused with an adapted two-step collagenase protocol as described (Wiechert et al, 2012; Klingmüller et al, 2006). The

animals were anesthetized with the anesthesia cocktail comprising 100 mg/kg body weight KETASET (Zoetis) and 10 mg/kg body weight of Rompun® 2% (Bayer). The abdominal cavity was opened to expose the vena portae. An Abbocath-T 24 G ¾ cannula (Megro GmbH & Co. KG) was inserted into the portal vein and perfused with perfusion buffer I (120 mM NaCl, 240 mM NaHCO$_3$, 20 mM Glucose, 5 mM HEPES, 4.8 mM KCl, 1.2 mM MgSO$_4$, 1.2 mM KH$_2$PO$_4$, 0.5 mM EGTA, 2% [v/v] Heparin 5000 U/ml, pH 7.4) pre-warmed at 42 °C with pump velocity at 2 ml/min for 10 min. Vena cava was incised to increase blood pressure and permit sufficient outflow. The perfusion was continued with buffer II (67 mM NaCl, 6.7 mM KCl, 20 mM CaCl$_2$, 100.7 mM HEPES and 0.1% [w/v] Collagenase type IV [Sigma], pH 7.6) with pump velocity at 4 ml/min for further 10 min.

## Isolation of non-parenchymal cells and FACS sorting for BECs

To isolate BECs, the perfused livers were resected and immersed in cold wash buffer (William's E medium [Life Technologies] supplemented with 10% FBS and 1% L-Glutamine). Liver capsules were opened up by forceps to release perfused cells. Cells in wash buffer were centrifuged at $300 \times g$ for 7 min at 4 °C. Cell pellet was digested in digestion buffer (0.1% [w/v] Collagenase type IV [Sigma], 0.1% [w/v] Pronase [Sigma] and 1% [w/v] DNase I [Sigma] in HBSS solution) for 1 hr at 37 °C with agitation. After digestion, 1:1 volume of cold wash buffer was added to digested cells and filtered through a 70-µm cell strainer, followed by centrifugation at $40 \times g$ for 5 min at 4 °C. Supernatant were transferred to a new tube and 1:1 volume of cold wash buffer was added, followed by centrifugation at $300 \times g$ for 7 min at 4 °C. Supernatant was discarded and cold wash buffer was added to resuspend the pellet. Cell suspension was gently transferred to the top of the 20%/50% Percoll (Sigma) in PBS and centrifuged at $1400 \times g$ for 20 min at 4 °C without brake. The layer of non-parenchymal cells (NPCs) was visible close to the 20%/50% interphase. Debris and fat in the top layer were aspirated and the NPC layer was pooled gently. The pooled cells were washed in cold wash buffer and centrifuged at $300 \times g$ for 12 min at 4 °C. Cell pellet was resuspended in 1% BSA/PBS and proceeded via FACS by FACSAria cell sorter. Cell sorting was gated for tdTomato and GFP expression. tdTomato positive cells from tamoxifen-treated $R26^{Tom}$ $Hnf1b$-$CreER$ $Rage^{+/fl}$ animals and tdTomato and GFP double positive cells from tamoxifen-treated $R26^{Tom}$ $Hnf1b$-$CreER$ $Rage^{fl/fl}$ animals were sorted directly into RNA lysis buffer of RNeasy kit for RNA isolation. RNA was concentrated by a rotational vacuum concentrator (Christ, Martin GmBH & Co. KG; Oterode, Germany).

## RNA sequencing and computational methods

Total RNA of BECs isolated from CDE diet-challenged $R26^{Tom}$ $Hnf1b$-$CreER$ mice ($n = 4$) or $R26^{Tom}Hnf1b$-$CreER$ $Rage^{\Delta BEC}$ mice ($n = 4$) in an amount of 10–25 ng per sample were subjected for library preparation utilizing the Ultra Low RNA v4 protocol, and were sequenced using Illumina HiSeq 4000 sequencing system (paired-end 100 bp) at DKFZ Genomics & Proteomics Core Facility. The Nextflow-based nf-core RNA-Seq pipeline (https://github.com/nf-core/rnaseq) was used for the bioinformatics analysis. An aggregation of the bioinformatics workflow analysis was conducted by MultiQC v1.7 (Ewels et al, 2016). FASTQC was

used to determine quality of the FASTQ files. Subsequently, adapter trimming was conducted with Trim Galore v0.6.4 (https://www.bioinformatics.babraham.ac.uk/projects/trim_galore/). STAR v2.6.1 d aligner (Dobin et al, 2013) was used to map the reads that passed the quality control to the reference genome. The evaluation of the RNA-seq experiment was performed with RSeQC v3.0.1 (Wang et al, 2012) and read counting of the features (e.g., genes) with featureCounts v1.6.4 (Liao et al, 2014).

For differential expression analysis, the raw read count table resulting from featureCounts was processed with the R package DESeq2 v1.22.1 (Love et al, 2014). The comparison was made between the gene expression data of the RAGE knockout BECs from RAGE$^{\Delta BEC}$ mice and RAGE control from RAGE$^{+/\Delta BEC}$ mice. The adjusted $P$ value was calculated in the DESeq2 package with the Benjamini–Hochberg method. Genes were considered differentially expressed (DE) when the adjusted $P$ value was lower than 0.05 ($P$-adj. value < 0.05).

The data have been deposited in NCBI's Gene Expression Omnibus (Edgar et al, 2002) and are accessible through GEO Series accession number GSE275751.

Graphs were also produced in the RStudio v1.1.456 with R version 3.5.1 mainly using the R package ggplot2 v3.2.1. Volcano plot displayed the DE genes in log2 fold change against their adjusted $p$ value in form of $-\log10$. KEGG and REACTOME pathway analyses were performed with the gProfiler2 tool. The sample similarity heatmap in form of regularized logarithm (rlog) normalized gene counts was created using the edgeR v3.26.5R.

## Standard histology, IHC, and IF staining on mouse tissues

For standard histological and immunohistochemistry (IHC) analyses, livers were removed, fixed in 4% PFA in PBS overnight, embedded in paraffin and sectioned at 4 μm thick for standard Hematoxylin and Eosin (H&E), Picro-Sirius Red or IHC staining. Standard IHC was performed with primary antibodies, biotinylated secondary antibodies and amplified by VECTASTAIN® avidin-biotin complex (ABC) solution (Vector Laboratories) for chromogenic detection by DAB (3,3'-diaminobenzidine) Peroxidase Substrate Kit (Vector Laboratories).

For oil red O and immunofluorescence (IF) staining, livers were removed, fixed in 4% PFA/PBS for 2–4 h, rinsed with PBS, and cryoprotected by incubating in increasing concentration of sucrose (Sigma) solution (10%, 20% and 30%) in PBS. The livers were embedded in O.C.T. Tissue-Tek, frozen on dry ice and sectioned at 5 μm thickness for Oil red O staining or IF staining. For IF staining, unless otherwise mentioned, the liver tissues were blocked with 0.1% BSA in 1× PBS, then incubated overnight with primary antibodies, followed by incubation with secondary antibodies for an hour if applicable, and nuclei staining with Hoechst 33342 (Biomol) for 10 min. For Ki67 and αSMA IF staining, liver tissues were first permeabilized with 0.5% Triton X-100 in 1× PBS for one hour, followed by using the general IF staining protocol as mentioned.

For multiplex IF staining, HES1 signal was amplified by TSA Biotin System and detected by Cy3 conjugated streptavidin (Vector Laboratories) diluted 1:100 in TNB blocking buffer. Subsequently, FlexAble CoraLite® Plus 647 and FlexAble CoraLite® Plus 750 Antibody Labeling Kits (Proteintech) were used to label anti-CK19 and anti-Desmin rabbit antibodies respectively. The liver tissues were incubated with the fluorescently labeled anti-CK19 and anti-

Desmin antibodies diluted in TNB blocking buffer overnight at 4 °C, followed by nuclei staining with Hoechst 33342 (Biomol). All IF Images were acquired with Zeiss Axio Scan.Z1 Slide Scanner at ×10, ×20, or ×40 magnification.

## In situ hybridization

In situ hybridization for the detection of *Rage* RNA expression on FFPE liver tissue was performed using RNAscope 2.5 HD Brown kit according to manufacturer's instructions. RNAscope probe targeting Mus musculus *Rage* transcript variant 1 at exon 2–7 (Probe Mm-Rage-O3, Catalog no. 882411) was designed by ACD.

## Histopathological evaluation

A blinded histopathological evaluation was conducted by a certified veterinary pathologist, Dr. Tanja Poth. H&E and Picros Sirius Red stained slides were evaluated using a general nonalcoholic fatty liver disease (NAFLD) scoring system described previously (Kleiner et al, 2005; Liang et al, 2014; Brunt et al, 1999) and adapted for this study. Histological features including steatosis, inflammation and ductular reaction were evaluated based on H&E staining; fibrosis was evaluated based on Picro-Sirius Red staining. The histological features of lobular inflammation was graded based on the number of inflammatory foci per 400× field while portal inflammation was graded based on severity as described previously (Brunt et al, 1999; Kleiner et al, 2005). Three features of steatosis, including macrovesicular steatosis (large lipid droplet present in hepatocytes), microvesicular steatosis (small lipid droplets present in hepatocytes) and hepatocellular hypertrophy, were graded based on the percentage of the total area affected: 0 (< 5%), 1 (5–33%), 2 (34–66%), and 3 (> 66%).

## Cell culture

All cell lines were maintained in an incubator at 37 °C with 5% $CO_2$. Primary BECs isolated from a *Rage$^{fl/fl}$* mouse were established according to the method previously described (Strick-Marchand and Weiss, 2002) and cultured in William's E medium supplemented with 5% FBS (Sigma-Aldrich), 30 ng/ml IGF2 (Sigma-Aldrich), 20 ng/ml EGF (Corning), 10 μg/ml insulin (Sigma-Aldrich), 2 mM L-glutamine and 1% penicillin/streptomycin. Immortalized hepatic stellate cell MIM1-4HSC (gift from Wolfgang Mikulits; (Proell et al, 2005) was cultured in DMEM high glucose (Sigma-Aldrich) supplemented with 10% FBS, 2 mM L-Glutamine and 1% penicillin/streptomycin.

To generate *Rage* wildtype (WT) and *Rage* knockout (KO) BEC cell lines, primary BECs described above were transiently transfected with pMXpie plasmid that either carries an empty vector (EV) or Cre recombinase (Cre) using FuGENE HD transfection reagent (Promega). Subsequently, cells were subjected to FACS by BD FACSAria I cell sorter. BECs transfected with pMXpie-EV were bulk-sorted, whereas BECs transfected with pMXpie-Cre were sorted for GFP, which is linked to the deletion of the *Rage* locus.

Viral transduction was performed for stable expression of mCherry in MIM1-4HSC. HEK293T cell was transfected with pUltra-hot plasmid using PEI for lentivirus production. MIM1-4HSC was transduced with the lentivirus, followed by FACS for mCherry-positive selection by BD FACSAria I cell sorter at 48-h post-transduction.

## Co-culture and conditioned medium assays

For direct co-culture and flow cytometry analysis, BECs and MIM1-4HSC-mcherry cells were seeded in 1:1 ratio and cultured in William's E medium supplemented with 5% FBS (Sigma-Aldrich), 30 ng/ml IGF2, 20 ng/ml EGF, 10 µg/ml insulin, 2 mM L-glutamine and 1% penicillin/streptomycin for 4 days. The cells were trypsinized and labeled with LIVE/DEAD™ Fixable Aqua Dead Cell Stain (Invitrogen), followed by fixation and permeabilization with BD Cytofix/Cytoperm™ Fixation/Permeabilization Solution Kit (BD Bioscience). The cells were stained with αSMA-eFluor660 antibody (eBioscience™) diluted at 1:50 in BD Perm/Wash™ buffer for 30 min prior to analysis with BD LSRFortessa™ cell analyser and FlowJo v10 Software.

For the conditioned medium assay, the conditioned medium from either Rage WT or Rage KO BECs was collected after 48 h of incubation in William's E cell culture medium, followed by centrifugation at 250 × g to remove debris. The supernatant was collected and filtered through a 0.22 µm filter. MIM1-HSC cells were seeded onto a 6-well plate or 24-well glass bottom plate (Cellvis) and incubated overnight prior to treatment with 5 ng/ml recombinant human TGFB1 protein (Bio-Techne) or BEC-conditioned medium (CM) for 48 h for RNA isolation, or 96 h for IF staining, respectively.

## Jagged1 induction assay

Recombinant Rat Jagged1 Fc Chimera Protein (Bio-Techne) was diluted in PBS to a final concentration of 1 µg/ml. The solution was placed onto a 24-well glass bottom plate (Cellvis) and incubated in a 37 °C incubator for 2 h. PBS was used as a control. The recombinant Jagged1 solution was aspirated and the plate was rinsed once with PBS. MIM1-4HSC cells were seeded on the Jagged1-coated 6-well plate and cultured in DMEM medium supplemented with 10% FBS, 2 mM L-glutamine, and 1% penicillin/streptomycin for 96 h in a 37 °C incubator for subsequent IF staining.

## Jagged1 silencing by RNAiMAX transfection

Rage WT BECs were seeded onto a six-well plate in William's E medium supplemented with 5% FBS, 30 ng/ml IGF2, 20 ng/ml EGF, 10 µg/ml insulin, 2 mM L-glutamine and 1% penicillin/streptomycin. Cells were incubated at 37 °C incubator overnight. The cell culture medium was refreshed prior to siRNA transfection. To transfect BECs with siRNA, lipofectamine® RNAiMAX reagent (Invitrogen) diluted with OptiMEM™ reduced serum medium (Gibco) was used. The Silencer™ Select Negative Control No. 1 siRNA (Invitrogen) and two anti-Jagged1 siRNA (Invitrogen; siRNA#1 ID s68530, siRNA#2 ID s68532) of 10 µM stock solution were mixed with OptiMEM serum free medium. After 5 min of incubation at room temperature, the siRNA-OptiMEM mix was added to RNAiMAX transfection reagent-OptiMEM mix in 1:1 ratio. After gentle mixing by pipetting, the siRNA-lipid complex mixture was added dropwise onto BECs and mixed by gentle rotation briefly. The final siRNA used per well was 25 pmol. Medium was replenished 6 h after transfection. At 72 h after siRNA transfection, the conditioned medium was collected, followed by debris removal via centrifugation at 250 × g and filtering through a 0.22-µm filter. The transfected BECs were subjected to RNA isolation and qPCR analysis.

## Quantitative protein analysis by mass spectrometry

Protein concentration of collected BEC- growth medium and conditioned medium were measured by Bradford assay. Protein samples were fractionated by SDS-PAGE followed by In-gel digestion. Resulting peptides were loaded on a cartridge trap column, packed with Acclaim PepMap300 C18, 5 µm, 300 Å wide pore (Thermo Scientific) and separated via a gradient from 3 to 40% ACN on a nanoEase MZ Peptide analytical column (300 Å, 1.7 µm, 75 µm × 200 mm, Waters) using a 60 min MS-method. Eluting peptides was analyzed by an online-coupled Orbitrap Exploris 480 mass spectrometer. Data analysis was carried out by MaxQuant (version 1.6.14.0). The match between runs option was enabled to transfer peptide identifications across Raw files based on accurate retention time and $m/z$. Quantification was done using a label-free quantification approach based on the MaxLFQ algorithm (Cox et al, 2014).

## Enzyme-linked immunosorbent assay (ELISA)

Rat Jagged1 ELISA Kit (Invitrogen) was utilized to analyze the concentration of Jagged1 in BEC-derived conditioned medium. BEC plain growth medium (William's E medium supplemented with 5% FBS, 30 ng/ml IGF2, 20 ng/ml EGF, 10 µg/ml insulin, 2 mM L-glutamine and 1% penicillin/streptomycin) and conditioned medium were diluted 1:30 in 1× assay diluent and added to rat Jagged1 antibody pre-coated wells for 2 h and 30 min of incubation at room temperature with gentle shaking. The solution was discarded, and the wells were washed four times with 1× wash buffer. Biotin conjugate was diluted 1:80 in 1× assay diluent and added to each well for 1 h of incubation at room temperature with gentle shaking. Following washing, the wells were incubated with streptavidin-HRP diluted 1:200 in 1× assay diluent for 45 min at room temperature with gentle shaking. After thoroughly washing, each well was incubated with TMB substrate for 30 min at room temperature in the dark with gentle shaking. Stop solution was added to each well and gently mixed to stop the reaction. The absorbance of the samples was measured at 450 nm with the CLARIOstar microplate reader (BMG LABTECH).

## Immunofluorescence staining on cells

Cells were fixed with 2% paraformaldehyde in PBS for 20 min, followed by washing with PBS for four times. Subsequently, cells were permeabilized by 0.2% triton-X in PBS for 30 min, and blocked with 10% goat serum in permeabilization buffer for 1 h at room temperature with gentle shaking. Cells were incubated with anti-αSMA primary antibodies and BODIPY™ 493/503 fluorescent dye diluted in blocking buffer overnight at 4 °C. Following washing with PBS, cells were incubated with Alexa Fluor 647 goat anti-rabbit secondary antibody diluted in blocking buffer for 1 h, and stained with Hoechst 33342 for 15 min. Finally, cells were washed with PBS and immersed in PBS for imaging. Immunofluorescence images were taken by confocal microscopy ZEISS LSM 710.

## Real-time PCR analysis

For cell culture, RNA was isolated using the RNeasy Mini Kit (Qiagen) according to the manufacturer's instructions. For mouse

tissues, livers were harvested, cut into pieces and snap-frozen. Tissues were grinded in an Eppendorf tube on ice and RNA was extracted using the RNeasy Mini Kit (Qiagen) according to the manufacturer's instructions. Following reverse transcription, qPCR was performed using Power SYBR™ Green PCR-Master Mix on a StepOnePlus™ Real-Time PCR System (Applied Biosystems™). Each sample comprises of three biological replicates tested in triplicates. The raw mean Ct value of each sample was normalized to the corresponding geometric mean Ct values of housekeeping genes *Hprt*. Sequences of primers used for the analysis are available in Appendix Table S2.

## Statistical test

Data were presented as mean value ± standard deviation (s.d.). Two-tailed *t* test was used for two groups and two-way ANOVA Turkey's multiple comparisons test was used for multiple groups of statistical comparison. Statistical tests were performed with GraphPad Prism 7.03. A *P* value < 0.05 with 95% confidence interval was considered statistically significant (*$P < 0.05$, **$P < 0.01$, ***$P < 0.001$, ****$P < 0.0001$).

# Data availability

The datasets produced in this study are available by deposition in NCBI's Gene Expression Omnibus (Edgar et al, 2002) and are accessible through GEO Series accession number GSE275751.

The source data of this paper are collected in the following database record: biostudies:S-SCDT-10_1038-S44319-024-00356-7.

# Peer review information

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

## Acknowledgements

The authors thank Angela Funk and Daniela Heide for the technical support, Wolfgang Mikulits for providing an immortalized hepatic stellate cell line, Manuela Brom from the DKFZ Core Facility "Light Microscopy" for assistance with the slide scanner and confocal microscope, Dr. Damir Krunic for assistance with image analysis, and the Center for Preclinical Research of DKFZ for animal care. The authors also like to thank Kai Breuhahn for advice during Macrina Lam's TAC meetings and Jan Hengstler, Nachiket Vartak, and Amruta Damle-Vartak for intensive discussions and for sharing unpublished data. This work was supported by the German Research Foundation (DFG) project number 314905040—CRC/SFB-TR 209 and GE 2289/3-1.

## Author contributions

**Wai-Ling Macrina Lam**: Conceptualization; Formal analysis; Investigation; Writing—original draft; Writing—review and editing. **Gisela Gabernet**: Software; Formal analysis. **Tanja Poth**: Supervision; Investigation. **Melanie Sator-Schmitt**: Investigation. **Morgana Barroso Oquendo**: Formal analysis; Investigation. **Bettina Kast**: Investigation. **Sabrina Lohr**: Investigation. **Aurora De Ponti**: Investigation. **Lena Weiß**: Investigation. **Martin Schneider**: Resources; Formal analysis; Investigation. **Dominic Helm**: Resources; Formal analysis; Investigation. **Karin Müller-Decker**: Resources; Supervision; Investigation. **Peter Schirmacher**: Resources; Project administration; **Mathias Heikenwälder**: Investigation. **Ursula Klingmüller**: Resources; Supervision. **Doris Schneller**: Conceptualization; Supervision; Funding acquisition; Investigation; Project administration. **Fabian Geisler**: Resources; Funding acquisition; Investigation; Writing—review and editing. **Sven Nahnsen**: Resources; Software; Supervision; Funding acquisition; Project administration. **Peter Angel**: Conceptualization; Formal analysis; Supervision; Funding acquisition; Investigation; Methodology; Writing—original draft; Project administration; Writing—review and editing.

Source data underlying figure panels in this paper may have individual authorship assigned. Where available, figure panel/source data authorship is listed in the following database record: biostudies:S-SCDT-10_1038-S44319-024-00356-7.

## Funding

## Disclosure and competing interests statement

The authors declare no competing interests.

# Expanded View Figures

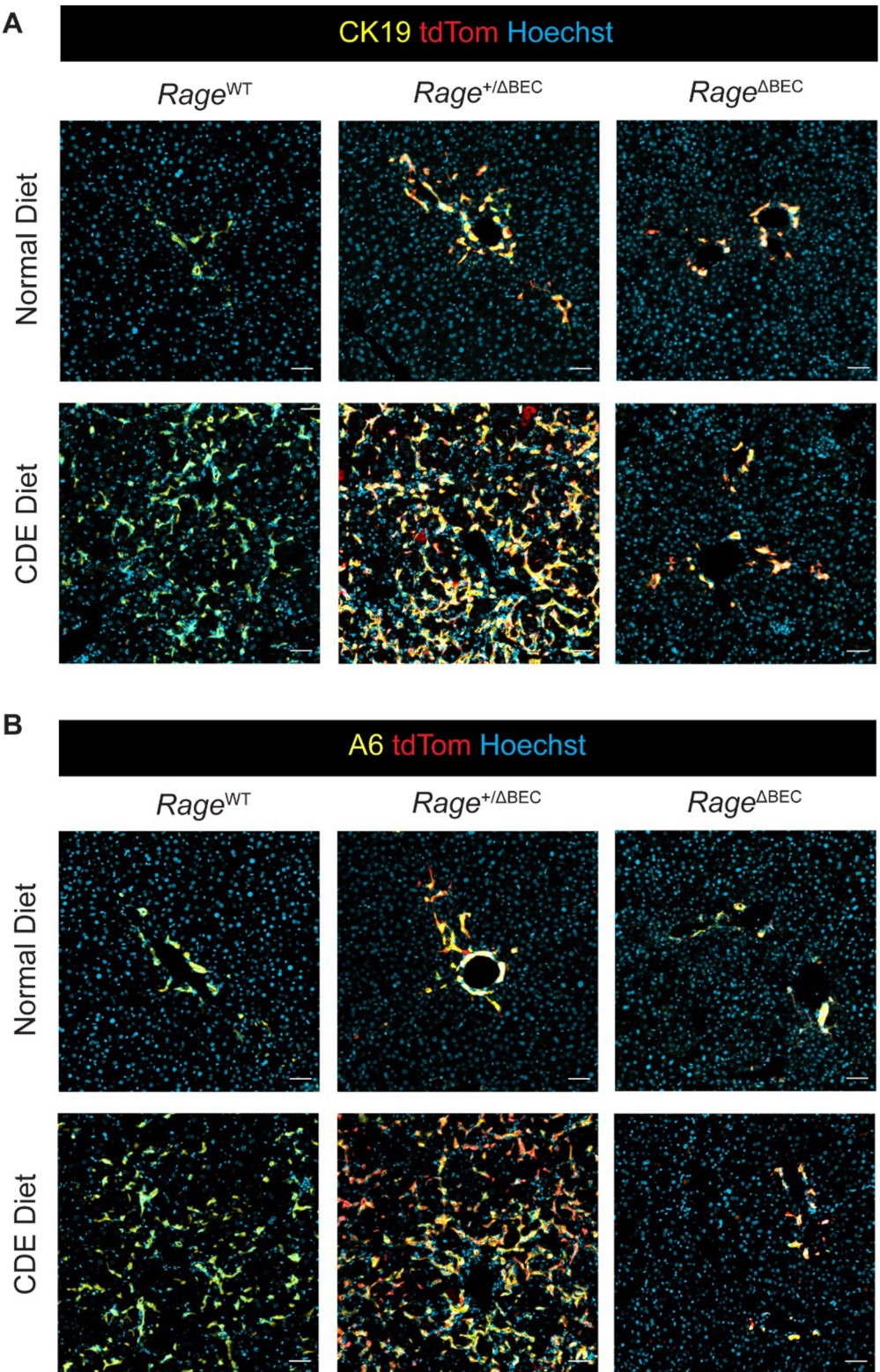

**Figure EV1. Co-staining of biliary markers and tdTomato-labeled biliary cells.**

(A) Representative images of IF of tdTomato and CK19. (B) Representative images of IF of tdTomato and A6. Scale bar = 50 μm. For normal diet (ND)-treated mice, $Rage^{WT}$ ($n = 12$), $Rage^{+/\Delta BEC}$ ($n = 12$), $Rage^{\Delta BEC}$ ($n = 11$); for CDE-treated mice, $Rage^{WT}$ ($n = 9$), $Rage^{+/\Delta BEC}$ ($n = 10$), $Rage^{\Delta BEC}$ ($n = 12$) (biological replicates).

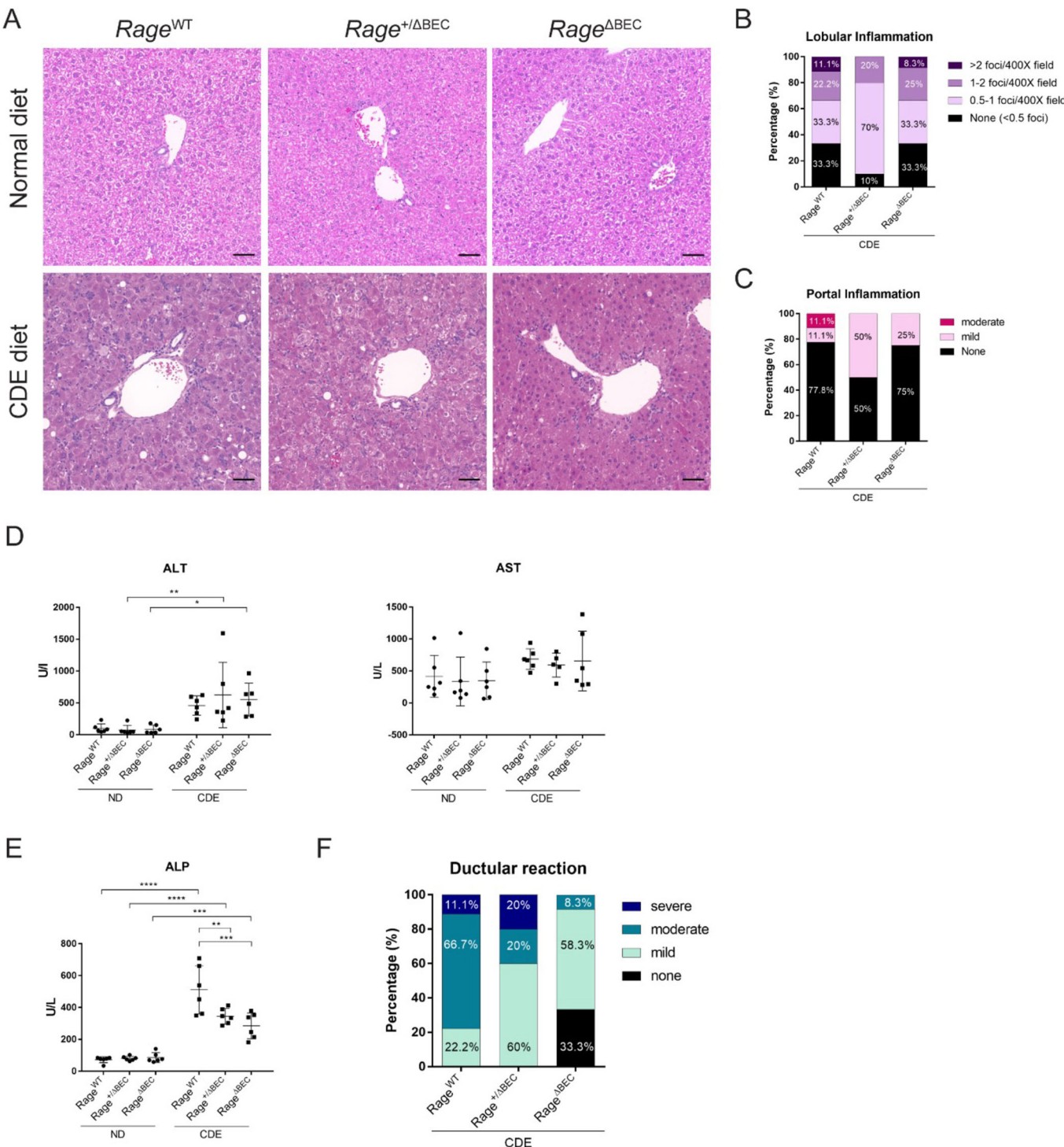

**Figure EV2. *Rage* in BEC is not involved in inflammation during cholestatic injury.**

(A) Hematoxylin & Eosin (H&E) staining of liver sections from *Rage*WT, *Rage*+/ΔBEC and *Rage*ΔBEC mice fed with normal or CDE diet for 3 weeks. Scale bar = 50 μm. (B, C) Histopathological evaluation of lobular inflammation and portal inflammation based on H&E staining in CDE diet-challenged mice. (D) Biochemical serum analysis of hepatic damage markers, alanine aminotransferase (ALT) and aspartate aminotransferase (AST) and (E) the marker of cholestasis, alkaline phosphatase (ALP). Data was shown as mean ± s.d. of *n* = 6 biological replicates per group. Two-way ANOVA with Turkey's multiple comparisons test was used for statistical comparison (*P < 0.05, **P < 0.01, ***P < 0.001, ****P < 0.0001). (F) Histopathological evaluation of DR in *Rage*WT, *Rage*+/ΔBEC and *Rage*ΔBEC mice fed with CDE diet for 3 weeks. For normal diet (ND)-treated mice, *Rage*WT (*n* = 12), *Rage*+/ΔBEC (*n* = 12), *Rage*ΔBEC (*n* = 11); for CDE-treated mice, *Rage*WT (*n* = 9), *Rage*+/ΔBEC (*n* = 10), *Rage*ΔBEC (*n* = 12).

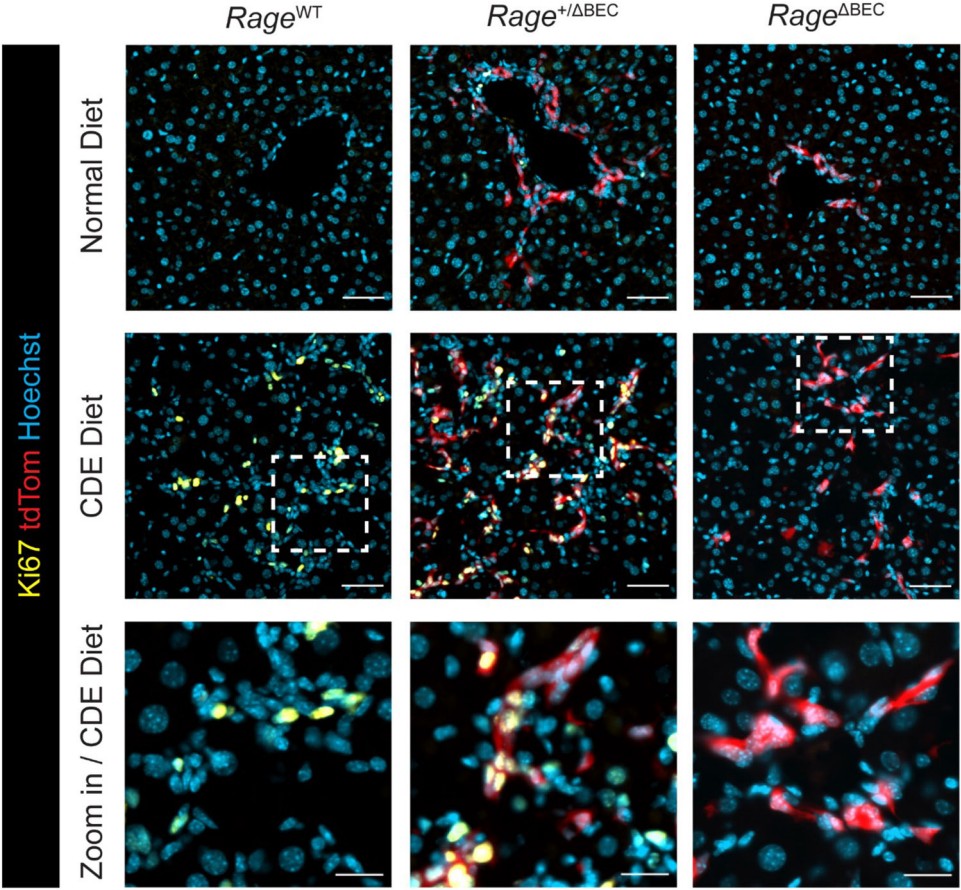

**Figure EV3.  *Rage* in BECs may contribute to BEC proliferation during chronic injury.**

IF Staining of proliferation marker Ki67 on liver sections from *Rage*^WT, *Rage*^+/ΔBEC and *Rage*^ΔBEC mice fed with normal or CDE diet for 3 weeks. At least $n = 3$ animals (biological replicates) were evaluated per group. Scale bar $= 50$ μm for top and middle row. Scale bar $= 20$ μm for bottom row.

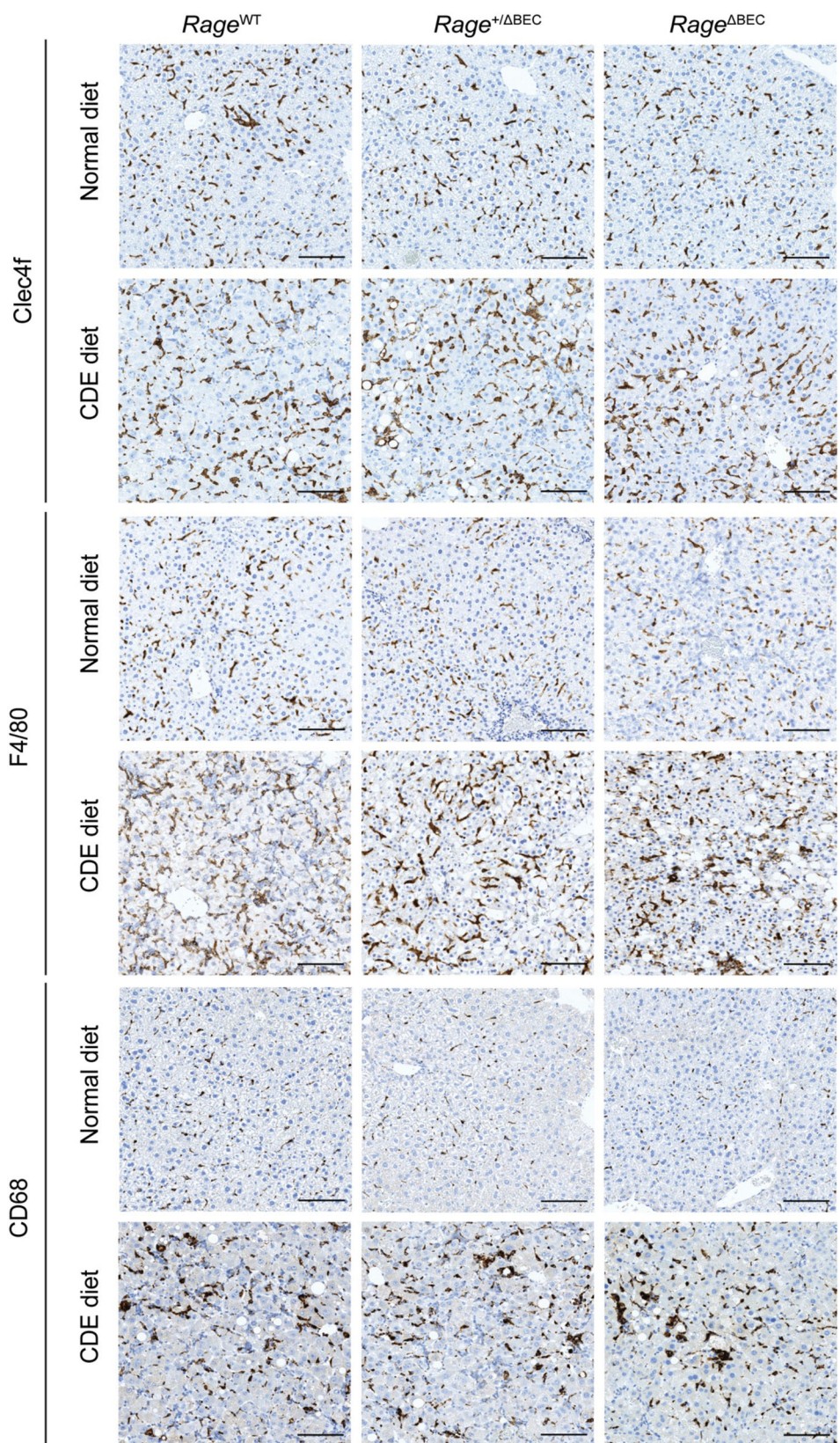

**Figure EV4. *Rage* in BECs does not contribute to immune cell infiltration during cholestatic injury.**

IHC staining of immune cells, including Clec4f for Kupffer cell, F4/80 for macrophages and CD68 for monocytes in *Rage*<sup>WT</sup>, *Rage*<sup>+/ΔBEC</sup> and *Rage*<sup>ΔBEC</sup> mice fed with normal or CDE diet for 3 weeks. At least $n = 3$ animals (biological replicates) per group were evaluated. Scale bar = 100 μm.

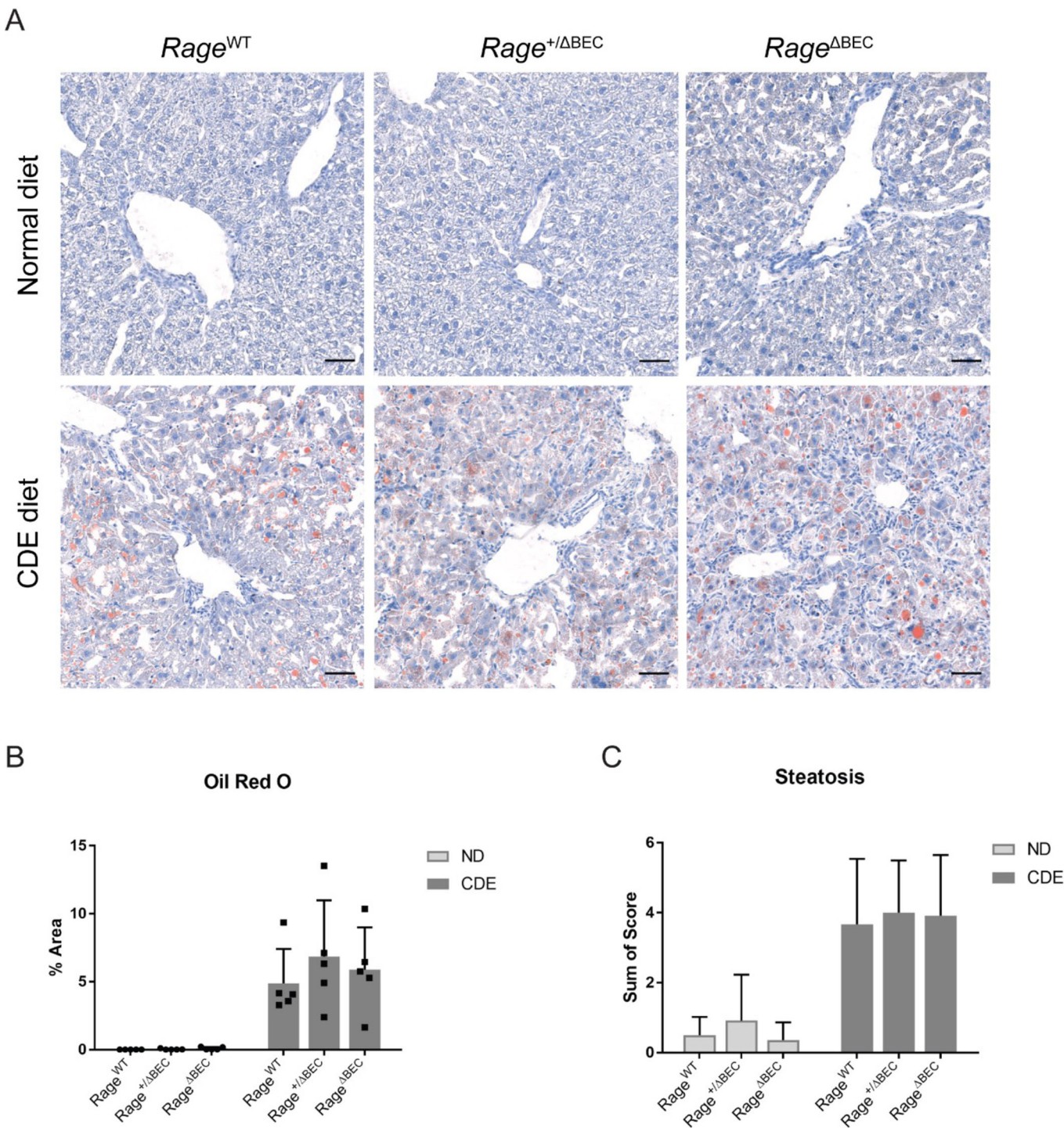

**Figure EV5. *Rage* in BEC is not associated with steatosis upon chronic liver injury.**

*Rage*WT, *Rage*+/ΔBEC and *Rage*ΔBEC mice were fed with normal or CDE diet for 3 weeks. (A) Representative images of Oil Red O staining (scale bar = 50 μm) and (B) corresponding Oil Red O quantification. Data was shown as mean ± s.d. of $n = 5$ animals (biological replicates) per group. Two-way ANOVA with Turkey's multiple comparisons test was used for statistical comparison. (C) Histopathological evaluation of steatosis (based on H&E staining in Fig. EV2A). Two-way ANOVA with Turkey's multiple comparisons test was used for statistical comparison.

