## [Peer Review File · EMBO Reports]

RAGE is a key regulator of ductular reaction-mediated fibrosis during cholestasis

Wai Ling Macrina Lam, Gisela Gabernet, Tanja Poth, Melanie Sator-Schmitt, Morgana Oquendo, Bettina Kast, Sabrina Lohr, Aurora De Ponti, Lena Weiß, Martin Schneider, Dominic Helm, Karin Mueller-Decker, Peter Schirmacher, Mathias Heikenwälder, Ursula Klingmüller, Doris Schneller, Fabian Geisler, Sven Nahnsen, and Peter Angel

Corresponding author(s): Peter Angel (P.Angel@dkfz.de)

Review Timeline:

Submission Date:	28th Feb 24
Editorial Decision:	9th Apr 24
Revision Received:	26th Oct 24
Editorial Decision:	26th Nov 24
Revision Received:	9th Dec 24
Accepted:	13th Dec 24

Editor: Esther Schnapp

Transaction Report:

Dear Prof. Angel,

Thank you for your patience while your manuscript was peer-reviewed at EMBO reports. We have now received the full set of referee comments that is pasted below.

As you will see, the referees acknowledge that the findings are potentially interesting. However, they also have some more suggestions for how the study could be improved, and several of these are overlapping among the reports. I think all suggestions are reasonable and should be addressed, but please let me know in case you disagree, and we can discuss the exact revisions requirements further, also in a video chat, if you like.

I would thus like to invite you to revise your manuscript with the understanding that the referee concerns must be fully addressed and their suggestions taken on board. Please address all referee concerns in a complete point-by-point response. Acceptance of the manuscript will depend on a positive outcome of a second round of review. It is EMBO reports policy to allow a single round of major revision only and acceptance or rejection of the manuscript will therefore depend on the completeness of your responses included in the next, final version of the manuscript.

We realize that it is difficult to revise to a specific deadline. In the interest of protecting the conceptual advance provided by the work, we recommend a revision within 3 months (10th Jul 2024). Please discuss the revision progress ahead of this time with the editor if you require more time to complete the revisions.

- 1) A data availability section providing access to data deposited in public databases is missing. If you have not deposited any data, please add a sentence to the data availability section that explains that.
- 2) Your manuscript contains statistics and error bars based on $n=2$. Please use scatter blots in these cases. No statistics should be calculated if $n=2$.

3) We replaced Supplementary Information with Expanded View (EV) Figures and Tables that are collapsible/expandable online. A maximum of 5 EV Figures can be typeset. EV Figures should be cited as 'Figure EV1, Figure EV2' etc... in the text and their respective legends should be included in the main text after the legends of regular figures.

5) a complete author checklist, which you can download from our author guidelines <https://www.embopress.org/page/journal/14693178/authorguide>. Please insert information in the checklist that is also reflected in the manuscript. The completed author checklist will also be part of the RPF.

6) Please note that all corresponding authors are required to supply an ORCID ID for their name upon submission of a revised manuscript (<https://orcid.org/>). Please find instructions on how to link your ORCID ID to your account in our manuscript tracking system in our Author guidelines <https://www.embopress.org/page/journal/14693178/authorguide#authorshipguidelines>

- the name of the statistical test used to generate error bars and P values,
- the number (n) of independent experiments (please specify technical or biological replicates) underlying each data point,
- the nature of the bars and error bars (s.d., s.e.m.),
- If the data are obtained from $n < 3$, use scatter blots showing the individual data points.

I look forward to seeing a revised form of your manuscript when it is ready.

Best regards,
Esther,

Referee #1:

The manuscript by Ai-Ling Macrina Lam and colleagues entitled "RAGE is a key regulator of ductular reaction-mediated fibrosis during cholestasis" expands previous work, which suggested a role for RAGE in ductular reaction (DR) and fibrosis, by deleting RAGE specifically in biliary epithelial cells (BECs) using a HNF1bCreERT2 driver. Their BEC-specific RAGE deletion add important additional insights into the role of RAGE in DR-induced fibrosis and the multicellular dynamics behind this process. Moreover, the authors provide interesting mechanistic insights into how BECs activate hepatic stellate cells (HSCs). The manuscript is well written and the conclusions are well supported by the provided data. However, there are some points that should be addressed before publication.

Major comments

- 1) The authors should add magnified insights in Figure 1C and add a costaining for a biliary marker to show selective tdTom/EGFP expression in biliary cells. At the current magnification and without an additional marker it is difficult to appreciate the BEC-specific recombination.
- 2) The HNF1b staining in Figure 1D suggests its upregulation in periportal hepatocytes following CDE diet. Do these hepatocytes also upregulate RAGE and is it still expressed since HNF1b-CreERT wasn't active in these cells? This would be a key differentiator from previous work using a full-KO for RAGE. Since both hepatocyte-to-BEC reprogramming as well as BEC proliferation contribute to a DR, this would add great value by disentangling these 2 RAGE-expressing populations.
- 3) The authors show that BEC-specific RAGE deletion does not impact inflammation. This makes sense given that CDE damages hepatocytes away from BECs and making up the majority of the space in the liver lobule. How does inflammation look like around BECs +/- RAGE deletion in the CDE model? Zoom-in images and additional/broader immune cell markers covering more than just macrophages (e.g. CD45) should be added, ideally in IFL costaining with EGFP or tdTOM with a biliary marker.
- 4) To further substantiate the BEC-induced HSC activation the authors should quantify the Desmin or Vimentin staining in relation to the distance from tdTOM+ or EGFP+ cells.
- 5) The JAG1 data is very interesting. While soluble JAG1 has been described, the major route for cell-cell communication is one cell expressing extracellular JAG1 attached to its membrane and a neighboring cell expression the Notch ligand. Can the authors show (or at least comment/speculate on) the contribution of both JAG1-related modi operandi in their model?

Minor comments

- 1) SuppFig2+4 shows no difference in steatosis, whereas a clear increase in steatosis is visible in Figure 3A. The authors should select more presentative images reflecting their quantifications.
- 2) Why are BECs in RAGE+/ΔBEC mice yellow? Wouldn't recombination in one allele be sufficient to induce EGFP expression?

Referee #2:

In this study the authors use a specific RAGE KO mouse model in BECs to investigate the impact of ductular reaction in CDE-induced fibrosis. The in vivo results show that RAGE in BECs is necessary for DR expansion and fibrosis accumulation, which is a very interesting finding. Moreover, they prove in vitro that BECs produce Jag1 in a RAGE-dependent manner, which acts as a paracrine signal to activate HSC through Notch signalling and induce fibrosis. However, in vitro system may not reflect what is seen in vivo. In vivo, RAGE deficiency has a profound effect on BECs, reducing the number and their profibrogenic profile, thus, it is difficult to discriminate these effect on BECs to the potential paracrine effect on HSCs via Jag1. The authors only evaluate the paracrine interaction of BECs with HSCs in vitro, where the number of BECs and their profile may not be as different as the one described in vivo.

Specific comments:

Which animals are used as RAGE^{WT}? Are R26Tom Hnf1b-CreER without crossing with RAGE fl/fl so the Cre background is common to all groups? Please specify in the methods section.

F1. Is this reduction in DR a result of decreased proliferation of BECs, increased apoptosis or senescence? Staining and quantification of HNF1b+ Ki67+, HNF1b+ cCasp3+ and HNF1b+ p21+ cells for instance may answer this question.

F1D should be quantified, Apart from in situ hybridization, it would be nice to have a quantification of RAGE at protein level (i.e. IHC or western blott of BECs vs non-BEC cells).

Regarding RAGE-specific deletion involvement in inflammation, it is true that the number of macrophages and histological evaluation between conditions does not seem to change. Quantification of stained area in all mice would provide more robust

results. Moreover, it would be relevant to assess the expression of pro-inflammatory cytokines at RNA level (i.e. CXCL-, TNF, IL1b, IL6...) and the infiltration of other immune cell types such as lymphocytes or neutrophils.

Figure 2C. How is the level of expression of mature cholangiocyte as well as immature/progenitor or stem markers?

Figure 2F, the expression of mesenchymal markers in BEC wt is remarkable, and may suggest increase EMT. Authors should evaluate the expression of EMT associated genes in the transcriptomic data. Also, BEC wt cells show a highly fibrogenic phenotype, which may be the responsible for the production of ECM. This should be discussed.

F3 a confirmation method such as hydroxyproline assay may be necessary.

F4A: Desmin and vimentin are not markers of activated HSCs as they are also expressed in quiescent HSCs, additional aSMA should be evaluated. Number of activated HSCs should be quantified.

F4A & SF 6: Higher magnification images should be provided to be able to evaluate the distribution of DR and HSC cells. It cannot be appreciated the statement of the authors "Of note, tdTom expression and Desmin or Vimentin staining were mutually exclusive but adjacent to each other within the liver parenchyma". Moreover, based on the images it cannot be concluded that "signifying a cell-cell interaction between BECs and HSCs during liver injury and fibrosis." Authors should rephrase the statement.

F4E. The effect of the RAGE deletion in the coculture is modest. Authors showed significant differences, statistics used in this graph should be explained. What is the number of replicates used?

F5: It is not clear if these studies were performed in freshly isolated quiescent HSCs. Based on the morphology cells seem already activated, and it is difficult to observe differences in activation in cells already activated in culture. aSMA should be quantified.

F6: It is surprising not to see aSMA in Control treated HSCs, as based on the nuclei it seems a confluent culture in vitro, which usually show a degree of activation. Can you please explain?

Why was Jag 1 selected. Genes such as TGFb are also differentially expressed in wt vs RAGE ko BEC cells which may have a profound effect on activation.

Is the phenotype of BECs immortalized cell line similar to the observed in vivo?

F8: To further strength results and prove that Rage-induced notch signaling is determining HSC activation, the authors should quantify the percentage of Desmin+Hes1+ cells. Assessing at protein levels some players of Notch signaling by western blot in the HSC isolated fraction could also confirm these results. Why is HES1 expression maintained in BEC?

Minor comments:

-Why Rage+/ Δ BEC already show a different phenotype from RageWT if in theory no changes in Rage levels are detected and if the Cre background is shared with RageWT (i.e. ALP levels, histopathological evaluations)? Is only one allele of Rage enough to full functionality?

-Why if DR participates as an escape route for accumulative bile during cholestasis, as you mention in the introduction, in this study the cholestasis marker ALP is reduced when there is fewer DR? May it depend on the model that is used? (i.e. DDC, BDL)

-Supplementary Figure 1B. It would be nice to add the control group just in case Rage specific expression changes upon injury.

-Apart from HSC activation, did they proliferate after direct co-culture with BECs? Was the total number of MIM1-4HSC-mch increased when cultured with the Rage ko condition? So it may not only be a matter of activation but also of increased number of HSC producing collagen.

Referee #3:

The paper investigates the role of expression of Rage receptor in the biliary tree of mice fed a modified choline deficient diet and show that Rage expression increases and if its expression is genetically blocked ductular reactive cells, as well as liver fibrosis, are decreased. This result is not linked to changes in liver inflammation, as the level of inflammation seems unaffected, but rather to the reduction of some Rage-dependent factor able to influence hepatic stellate cell activation. The authors identify this factor as a secreted form of Jagged-1, a Notch ligand. The authors need to be congratulated for their extensive and careful work, however there are concerns that decrease the enthusiasm:

1) CDE diet is not "cholestatic" is a model of steatohepatitis. This needs to be corrected.

2) Thus, the biliary tree in this condition probably plays a reparative function, rather than an inflammatory one.

3) Only gene expression is shown for Rage- It would be important to show it at the protein level.

4) It is not clear how the authors envision the effects of Trage on DR cells expansion: direct proliferation of cells, or a stimulation from hepatic stellate cells stimulated by BDE-secrete4d Jagged-1?

5) Page 13: 'Under normal conditions, BECs line the bile duct at the biliary compartment of the portal vein (Supplementary Figure 1A).' I am not sure what this means, unless what this is supposed to mean, unless the authors are referring to the portal space.

6) Page 16: "we isolated primary BECs from CDE diet-treated Ragefl/fl mouse to establish a Rage knockout cell line" this is unclear- was this isolation done after administration of tamoxifen?

7) The authors find jagged-1 on the supernatant of BDE- soluble Jagged-1 inhibits notch signaling, whereas jagged immobilized to the plasticware (as they have correctly done) stimulates Notch. What mechanisms do the author envision? A cell-to-cell interaction or a paracrine mechanism?

Reviewer' Comments:

Reviewer 1:

The manuscript by Ai-Ling Macrina Lam and colleagues entitled "RAGE is a key regulator of ductular reaction-mediated fibrosis during cholestasis" expands previous work, which suggested a role for RAGE in ductular reaction (DR) and fibrosis, by deleting RAGE specifically in biliary epithelial cells (BECs) using a HNF1bCreERT2 driver. Their BEC-specific RAGE deletion add important additional insights into the role of RAGE in DR-induced fibrosis and the multicellular dynamics behind this process. Moreover, the authors provide interesting mechanistic insights into how BECs activate hepatic stellate cells (HSCs). The manuscript is well written and the conclusions are well supported by the provided data. However, there are some points that should be addressed before publication.

Major comments

1) The authors should add magnified insights in Figure 1C and add a costaining for a biliary marker to show selective tdTom/EGFP expression in biliary cells. At the current magnification and without an additional marker it is difficult to appreciate the BEC-specific recombination.

We have performed additional co-stainings for CK19/tdTom as well as A6/tdTom, which is now presented in Appendix Figure 2. These data clearly show that tdTomato is co-expressed selectively in CK19+ and A6+ BECs underscoring that Cre activity (and, thus, *rage* deletion) is present exclusively in BECs.

2) The HNF1b staining in Figure 1D suggests its upregulation in periportal hepatocytes following CDE diet. Do these hepatocytes also upregulate RAGE and is it still expressed since HNF1b-CreERT wasn't active in these cells? This would be a key differentiator from previous work using a full-KO for RAGE. Since both hepatocyte-to-BEC reprogramming as well as BEC proliferation contribute to a DR, this would add great value by disentangling these 2 RAGE-expressing populations.

Hepatocytes have no to very low expression of RAGE. The exact contribution of hepatocyte reprogramming into cholangiocytes in DR formation is still a matter of debate. In a study utilizing BEC lineage tracing with Hnf1b-CreER mouse, it was reported that DRs primarily arise from the biliary compartment (Jörs S et al., JCI 2015; doi: [10.1172/JCI78585](https://doi.org/10.1172/JCI78585)). Based on the findings of this article and multiple others, hepatocytes do not reprogram into cholangiocytes during liver regeneration in the CDE model where DRs/ductular reactive BECs entirely arise from the biliary compartment. Robust hepatocyte-to-BEC reprogramming occurs in primarily cholestatic models like the BDL model and, to a smaller extent, in the DDC model (There are multiple excellent studies on this i.e., Yanger 2013, Genes Dev 27:719-24).

3) The authors show that BEC-specific RAGE deletion does not impact inflammation. This makes sense given that CDE damages hepatocytes away from BECs and making up the

majority of the space in the liver lobule. How does inflammation look like around BECs +/- RAGE deletion in the CDE model? Zoom-in images and additional/broader immune cell markers covering more than just macrophages (e.g. CD45) should be added, ideally in IFL costaining with EGFP or tdTOM with a biliary marker.

In addition to Clef4f and F4/80, we have quantified the abundance of monocytes using CD68 as a marker. These data are now also included into Appendix Figure S5. Again, no obvious differences were detected. Moreover, I like to mention yet unpublished data from the close collaboration of the Angel and Geisler labs with the lab of Jan Hengstler on the function of RAGE in BECs on bile acid transport and biliary drainage using the BEC-specific RAGE KO system used in our present study. The corresponding manuscript (Damle-Vartak, ...Lam, ..., Angel, Hengstler, Vartak and Geisler; 2024), which is currently under review contains data on cytokine profiling of liver tissues from these mice. No inflammation-related signature of genes could be identified, whose expression differ in KO mice as compared to the Het controls. Although we cannot rigorously exclude subtle changes in defined immune cell subtypes, we feel confident to exclude RAGE in BECs as a major regulator of immune cell infiltration in our CDE model.

4) To further substantiate the BEC-induced HSC activation the authors should quantify the Desmin or Vimentin staining in relation to the distance from tdTOM+ or EGFP+ cells.

To get quantitative data turned out to be extremely challenging and did not yield convincing results. Instead, we followed the suggestion of the other reviewer to include additional stainings using α SMA as a marker for activated HSCs. These data are now included into Figure 4 (C). These data show that only in RAGE^{WT} and RAGE^{+/ Δ BEC} mice, but not in RAGE ^{Δ BEC} mice activated α SMA-positive cells are clearly visible in the liver parenchyme in close proximity to BECs.

5) The JAG1 data is very interesting. While soluble JAG1 has been described, the major route for cell-cell communication is one cell expressing extracellular JAG1 attached to its membrane and a neighboring cell expression the Notch ligand. Can the authors show (or at least comment/speculate on) the contribution of both JAG1-related modi operandi in their model?

We agree with this reviewer that both options are very well valid to execute JAG1 function on HSCs. On the one hand, we showed that recombinant JAG1 immobilized to the plastic ware can activate HSCs in vitro. Nevertheless, because the concentration of soluble Jag in the supernatant from BEC RAGE^{WT} or KO cells is very low, upon immobilization to the plate it remains uncertain whether this experimental setup mimics the “direct cell-cell interaction” environment. Despite the limitation of our in vitro model, and considering, on the one hand the close proximity of BECs and α -SMA-positive HSCs in vivo, and, on the other hand, soluble JAG1 to be able to activate HSCs via paracrine signaling in our cell culture model, we speculate that in real world scenario, chronic injury in the liver triggers BECs proliferation and JAG1 expression, resulting in presentation of the ligand on BEC’s

surface and binding to HSCs, which express the Notch receptor. We have changed the text in the Discussion accordingly mentioning both modes of action.

Minor comments

1) SuppFig2+4 shows no difference in steatosis, whereas a clear increase in steatosis is visible in Figure 3A. The authors should select more presentative images reflecting their quantifications.

We thank the reviewer for this suggestion. More representative images were now selected.

2) Why are BECs in *Rage*+/ Δ BEC mice yellow? Wouldn't recombination in one allele be sufficient to induce EGFP expression?

We assume that the reviewer is asking for an explanation why BECs in *Rage* Het mice are not yellow indicating EGFP expression (originating from one deleted *rage* allele). We do not have a very solid explanation but argue that targeting only one allele of the *Rage* locus (as confirmed by PCR analysis) is indeed not sufficient to yield GFP levels that are above detection threshold. This is also supported by the FACS sorting approach shown in Figure 2, where no GFP signals are visible in BECs from *Rage* Het mice.

Reviewer 2:

In this study the authors use a specific *Rage* KO mouse model in BECs to investigate the impact of ductular reaction in CDE-induced fibrosis. The in vivo results show that *Rage* in BECs is necessary for DR expansion and fibrosis accumulation, which is a very interesting finding. Moreover, they prove in vitro that BECs produce Jag1 in a *Rage*-dependent manner, which acts as a paracrine signal to activate HSC through Notch signaling and induce fibrosis. However, in vitro system may not reflect what is seen in vivo. In vivo, RAGE deficiency has a profound effect on BECs, reducing the number and their profibrogenic profile, thus, it is difficult to discriminate these effects on BECs to the potential paracrine effect on HSCs via Jag1. The authors only evaluate the paracrine interaction of BECs with HSCs in vitro, where the number of BECs and their profile may not be as different as the one described in vivo.

Specific comments:

Which animals are used as *Rage*WT? Are R26Tom *Hnf1b*-CreER without crossing with *Rage* fl/fl so the Cre background is common to all groups? Please specify in the methods section.

For the current model lacking RAGE in BECs we have used R26Tom *Rage* +/+ mice either lacking Cre (this study) or containing Cre (unpublished and used in the joint paper with the Hengstler lab described above) and no differences were seen on the level of DR formation or collagen deposition confirming that Cre expression does not

affect the processes analyzed in this study. Details on the mouse strains are now included into the methods section.

F1. Is this reduction in DR a result of decreased proliferation of BECs, increased apoptosis or senescence? Staining and quantification of HNF1b+ Ki67+, HNF1b+ cCasp3+ and HNF1b+ p21+ cells for instance may answer this question.

We have followed this suggestion by measuring proliferation of BECs in vivo via Ki67 staining. As now presented in Appendix Figure S3, a significant number of BECs (visualized by tdTom staining in Rage het mice) exhibit Ki67 signals, where in RAGE KO BECs no KI67 signals can be detected showing that RAGE deficiency strongly impairs BEC proliferation.

F1D should be quantified, Apart from in situ hybridization, it would be nice to have a quantification of Rage at protein level (i.e. IHC or western blott of BECs vs non-BEC cells). **We agree that quantification on the level of RAGE protein would be desirable. However, RAGE is hardly detectable on liver tissues or cell line. We have tested a variety of commercially available antibodies, which are supposed to specifically recognize murine RAGE. A valid Rage antibody (Abcam ab219326) was identified, which shows positive staining on lung tissue from a Rage WT (Rage +/fl) mouse and negative staining in Rage KO (Rage -/-) mouse (Figure panel A). However, RAGE was not detectable on liver tissues from normal diet- or CDE diet-treated mice (panel B). Furthermore, western blot also showed that Rage is only detectable in lung tissues but not in cultured liver BECs (Figure panel D). All other antibodies tested failed due to the lack of specificity (panel D).**

Regarding RAGE-specific deletion involvement in inflammation, it is true that the number of macrophages and histological evaluation between conditions does not seem to change. Quantification of stained area in all mice would provide more robust results. Moreover, it would be relevant to assess the expression of pro-inflammatory cytokines at RNA level (i.e. CXCL-, TNF, IL1b, IL6...) and the infiltration of other immune cell types such as lymphocytes or neutrophils.

As described above, in addition to Clef4f and F4/80, we have quantified the abundance of monocytes using CD68 as a marker. These data (upper panel of subsequent figure) are now also included into Appendix Figure S5. For your information, we have also quantified (but not included into the figure due to space restrictions) the abundance of Cd68 positive cells. Again, no obvious differences were detected.

Moreover, as described above, in the course of our collaboration with the Hengstler lab using the BEC-specific RAGE KO system used in our present study data on cytokine profiling of liver tissues from these mice were generated and are part of a submitted manuscript (Damle-Vartak et al). Here, no inflammation-related signature of genes could be identified, whose expression differ in KO mice as compared to the Het controls. Therefore, we feel confident to exclude RAGE in BECs as a major regulator of immune cell infiltration in our CDE model.

Figure 2C. How is the level of expression of mature cholangiocyte as well as immature/progenitor or stem markers?

Figure 2F, the expression of mesenchymal markers in BEC wt is remarkable, and may suggest increase EMT. Authors should evaluate the expression of EMT associated genes in the transcriptomic data. Also, BEC wt cells show a highly fibrogenic phenotype, which may be the responsible for the production of ECM. This should be discussed.

As shown below, expression of mature cholangiocyte marker KRT19 stays at the same level comparing Rage Co versus KO primary isolated BECs. Progenitor/epithelial markers such as EpCAM also show similar RNA levels between Rage Control and KO BECs, but progenitor/ mesenchymal markers such as CD44 (Ref: Yovchev et. al. Hepatology (2007). DOI: 10.1002/hep.21448) and Thy-1 (Ref.: Takase et. al. Genes & Dev (2013). DOI: 10.1101/gad.204776.112) were downregulated in Rage KO BECs, suggesting that the BECs are more epithelial-like and less mesenchymal in the absence of Rage activity.

Data on differentially expressed CD44 and Thy1 are now presented in Appendix Figure 11.

F3 a confirmation method such as hydroxyproline assay may be necessary.

We agree that , if feasible, hydroxyproline would be a good assay using liver tissue or serum. Unfortunately, only residual liver tissue and no serum is available anymore and there is no possibility to generate fresh material. On the other hand, we have additionally included imunohistochemical staining for aSMA positive cells in the parenchyme supporting the presence of activated stellate cells and formation of fibrosis.

F4A: Desmin and vimentin are not markers of activated HSCs as they are also expressed in quiescent HSCs. Number of activated HSCs should be quantified.

We have followed this suggestion and performed additional aSMA stainings. Indeed, aSMA-positive cells are clearly visible in WT and Hets in the vicinity of BECs but are clearly absent in the surrounding of the few BECs in RAGE-KO livers. These data are now included into the manuscript (Fig. 4) and text was changed accordingly.

F4A & SF 6: Higher magnification images should be provided to be able to evaluate the distribution of DR and HSC cells. It cannot be appreciated the statement of the authors "Of note, tdTom expression and Desmin or Vimentin staining were mutually exclusive but adjacent to each other within the liver parenchyma". Moreover, based on the images it cannot be concluded that "signifying a cell-cell interaction between BECs and HSCs during liver injury and fibrosis." Authors should rephrase the statement.

We have rephrased our data on Desmin and Vimentin expression accordingly and focused on the spatial distribution of aSMA-positive HSCs and K19-positive BECs.

F4E. The effect of the RAGE deletion in the coculture is modest. Authors showed significant differences, statistics used in this graph should be explained. What is the number of replicates used? .

Three replicates were used. The aSMA expression HSC were differentiated into aSMA low and high populations and compared among three co-culture conditions. Two way ANOVA Turkey's multiple comparison test was used for statistical comparison. This information is now given in the corresponding figure legend.

F5: It is not clear if these studies were performed in freshly isolated quiescent HSCs. Based on the morphology cells seem already activated, and it is difficult to observe differences in activation in cells already activated in culture.

The HSCs were originally isolated from p19(ARF) deficient model. Primary HSCs isolated from livers of adult p19 ARF null mice underwent spontaneous immortalization with signs of activation through long-term passaging without an obvious replicative limit (Proell et al 2005). The HSCs used in this study indeed exhibited signs for an at least partially activated status; upon treatment with rTGFB1 or BEC WT CM, aSMA expression in HSCs was further enhanced.

F6: It is surprising not to see aSMA in Control treated HSCs, as based on the nuclei it seems a confluent culture in vitro, which usually show a degree of activation. Can you please explain?

In this setting, the HSCs were indeed in a confluent state and we can only speculate that these HSC line in confluent culture will generally downregulate gene expression including aSMA expression, and, thus, cells show a lesser degree of aSMA signals.

Why was Jag 1 selected. Genes such as TGFB are also differentially expressed in wt vs RAGE ko BEC cells which may have a profound effect on activation.

The main reason for selecting JAG1 was the fact that TGFB1 did not show up in our mass spectrometry results of differential expression of soluble factors when

comparing BEC Rage WT vs KO condition medium, arguing that TGFB1 protein levels, despite differences in RNA abundance between in vivo isolated Control and KO BECs cells, are below detection levels in cultured BECs. Moreover, JAG1 was found to be a critical factor that determines biliary tree development in the liver, and deregulation in JAG1 leads to biliary tree paucity. Detecting major RAGE-dependent differences in JAG1 levels, we, therefore, speculate that JAG1 might a crucial factor contributing to BEC (DR)-induced fibrosis in our CDE model setting. Nevertheless, it is reasonable to assume that in vivo TGFb (and other previously identified cholangiokines) might still be essential for mounting the fibrosis phenotype. We have modified and completed this aspect in the Discussion section.

Is the phenotype of BECs immortalized cell line similar to the observed in vivo?

The BECs cell lines were isolated from CDE diet-treated Rage fl/fl mice and passaged in culture. Possibly due to the selection pressure no obvious differences in proliferation could be observed marking a major difference to the in vivo situation. On the other hand, gene expression patterns of BECs isolated in vivo were recapitulated in the immortalized BEC cell line for some genes, but not for others. Currently, there is no clear picture to draw a decisive conclusion on the degree of similarity between BECs in vivo and the BEC cell line cultured in vitro.

F8: To further strength results and prove that Rage-induced notch signaling is determining HSC activation, the authors should quantify the percentage of Desmin+Hes1+ cells. Assessing at protein levels some players of Notch signaling by western blot in the HSC isolated fraction could also confirm these results. Why is HES1 expression maintained in BEC?

Quantification turned out to be very challenging and did not yield robust results. It is important to note that Notch signaling was found to be also crucial for biliary tree development, thus exhibiting expression also in BECs. Assessing at protein levels some players of Notch signaling by western blot is indeed a very valid strategy; however, we do not have the possibility anymore (due to retirement) to perform additional in vivo experiments in order to freshly isolate HSCs from CDE-treated Wt and mutant mice.

Minor comments:

-Why Rage+/ Δ BEC already show a different phenotype from RageWT if in theory no changes in Rage levels are detected and if the Cre background is shared with RageWT (i.e. ALP levels, histopathological evaluations)? Is only one allele of Rage enough to full functionality?

In our previous work using total RAGE knockouts in skin and liver, looking at aspects of inflammation and tumorigenesis, heterozygous mice when tested always followed the phenotype of wildtype mice but not the mutants (reduced inflammation and tumorigenesis in skin; fibrosis and HCC in liver). Based on these data, one allele is sufficient for full functionality regarding the aforementioned phenotypes. For the

current model lacking RAGE in BECs we have used R26Tom RAGE +/- mice either lacking Cre (this study) or containing Cre (unpublished and used in the joint paper with the Hengstler lab) and no differences were seen on the level of DR formation. Obviously, also here one allele is sufficient for full functionality. Nevertheless, we cannot exclude that RAGE levels might be rate-limiting for expression of certain target genes and downstream affected processes might be or not be affected accordingly. Nevertheless, focusing on differences also seen when comparing one and null *Rage* alleles may identify the most critical targets of RAGE function.

-Why if DR participates as an escape route for accumulative bile during cholestasis, as you mention in the introduction, in this study the cholestasis marker ALP is reduced when there is fewer DR? May it depend on the model that is used? (i.e. DDC, BDL)

The mechanism, why AP rises in cholestasis is not well understood. Presumably it is released from sinusoidal vesicles and enhanced synthesis in canaliculi in response to cholestasis and canalicular damage. We can speculate that DR may be an ad hoc reparative attempt to provide an escape route – however it remains unclear at this point whether the net effect on biliary drainage of persistent DRs with its inflammatory niche is rather beneficial or detrimental (as is the case with any inflammatory/regenerative response in life). Moreover, it is not clear, whether or not ALP is a direct target of RAGE signaling and whether RAGE levels on BECs are rate-limiting for full ALP expression. Therefore, hypothesis on the reason for differential expression of ALP in the various genotypes remains speculative. Similarly, experimental data are needed to resolve the question whether or not our data are also true for the other models mentioned above.

-Supplementary Figure 1B. It would be nice to add the control group just in case *Rage* specific expression changes upon injury.

Unfortunately, in unchallenged mice, BECs represent only a very small population in the liver and, thus, it was very challenging (and not possible) to isolate a significant amount of BECs from control mice under normal condition. Therefore, *Rage* was only compared with Hepatocytes isolated from mice under chronic injury state.

-Apart from HSC activation, did they proliferate after direct co-culture with BECs? Was the total number of MIM1-4HSC-mch increased when cultured with the *Rage* ko condition? So it may not only be a matter of activation but also of increased number of HSC producing collagen.

We did not quantify proliferation of MIM1-4HSC-mch, e.g. via BrdU incorporation on a single-cell level via IF in mono- and co-cultures or in monocultures upon treatment with conditioned medium but according to microscopy there was no obvious difference.

Reviewer 3:

The paper investigates the role of expression of Rage receptor in the biliary tree of mice fed a modified choline deficient diet and show that Rage expression increases and if its expression is genetically blocked ductular reactive cells, as well as liver fibrosis, are decreased. This result is not linked to changes in liver inflammation, as the level of inflammation seems unaffected, but rather to the reduction of some Rage-dependent factor able to influence hepatic stellate cell activation. The authors identify this factor as a secreted form of Jagged-1, a Notch ligand. The authors need to be congratulated for their extensive and careful work, however there are concerns that decrease the enthusiasm:

1) CDE diet is not "cholestatic" is a model of steatohepatitis. This needs to be corrected.

We acknowledge this reviewer's remark and confidently state that of course, the CDE-model is not a classical "cholestasis model", as extrahepatic and intrahepatic bile ducts are primarily not affected. On the other hand, however, the CDE-model is a Hepatitis-model, which, without doubt, leads to intrahepatic cholestasis due do hepatocyte damage with disruption of the canalicular architecture/connectivity/function to Canals of Hering. We have modified the wording accordingly in the first part of the Discussion section.

2) Thus, the biliary tree in this condition probably plays a reparative function, rather than an inflammatory one.

As described above, we can only speculate that DR may be an ad hoc reparative attempt to provide an escape route – however it remains unclear at this point whether the net effect on biliary drainage of persistent DRs with its inflammatory niche is rather beneficial or detrimental (as is the case with any inflammatory/regenerative response). It is tempting to speculate that once unhinged, detrimental effects (such as fibrosis) may prevail.

3) Only gene expression is shown for Rage- It would be important to show it at the protein level.

I fully agree with this reviewer's concern. However, as pointed out already above, although tested extensively, there is no RAGE antibody available that allows detection of low RAGE levels in BECs. Therefore, in situ hybridization remains the only option to visualize Rage expression on tissue samples.

4) It is not clear how the authors envision the effects of rage on DR cells expansion: direct proliferation of cells, or a stimulation from hepatic stellate cells stimulated by BDE-secreted Jagged-1?

We speculated that there might be a positive feedback loop between BECs and HSCs, but the initial trigger originates from Rage-dependent activation and expansion of BECs. Rage activity in BECs is a critical factor that regulates BEC proliferation and expansion, thereby playing a feed-forward mechanism that induces HSC activation via Jagged1 to foster fibrosis in Rage WT setting. In the absence of Rage activity on BECs, this positive feedback loop is much less active.

5) Page 13: "Under normal conditions, BECs line the bile duct at the biliary compartment of the portal vein (Supplementary Figure 1A)." I am not sure what this means, unless what this is supposed to mean, unless the authors are referring to the portal space.

Indeed, this wording might be confusing and, for clarification we have replaced it accordingly to read "BECs line the extra- and intrahepatic biliary tree. Under homeostatic conditions, A6/CK19+ bile ducts/ductules are confined to the portal space in direct proximity to the portal vein."

6) Page 16: "we isolated primary BECs from CDE diet-treated *Rage*^{fl/fl} mouse to establish a *Rage* knockout cell line" this is unclear- was this isolation done after administration of tamoxifen?

The primary BECs were isolated separately from a CDE-treated *Rage* fl/fl (containing no Cre) omitting administration of tamoxifen. To generate *Rage* KO cell lines, Cre was introduced into these cells via viral infection to create genetic recombination and deletion of *Rage* exons 2-7 to abolish *Rage* function. Controls cells were infected by virus lacking Cre sequences.

7) The authors find jagged-1 on the supernatant of BDE- soluble Jagged-1 inhibits notch signaling, whereas jagged immobilized to the plasticware (as they have correctly done) stimulates Notch. What mechanisms do the author envision? A cell-to-cell interaction or a paracrine mechanism?

The reviewer may have misinterpreted our hypothesis and conclusion. We hypothesized that soluble Jagged 1 from BECs stimulate Notch signaling in HSCs, which can be demonstrated by both treatment with Jagged 1-containing supernatant of BECs and immobilization of Jagged1 on the plastic dish.

As pointed out above, both options, - a cell-to-cell interaction or a paracrine mechanism-, are very well valid to execute JAG1 function on HSCs. On the one hand, we showed that recombinant JAG1 immobilized to the plastic ware can activate HSCs in vitro. Nevertheless, because the concentration of soluble Jagged 1 in the supernatant from BEC *Rage* WT or KO cells is very low, upon immobilization to the plate it remains uncertain whether this experimental setup mimics the "direct cell-cell interaction" environment. Considering, on the one hand the close proximity of BECs and α -SMA-positive HSCs *in vivo*, and, on the other hand, soluble JAG1 to be able to activate HSCs via paracrine signaling in our cell culture model, we speculate that *in vivo* chronic injury in the liver triggers BECs proliferation and JAG1 expression, resulting in presentation of the ligand on BEC's surface and binding to HSCs (without excluding the function of a paracrine mechanism), which express the Notch receptor. We have changed the text in the Discussion accordingly mentioning both modes of action.

Dear Prof. Angel,

Thank you for the submission of your revised manuscript. We have now received the enclosed reports from the referees and I am happy to say that all support its publication now. Only a few editorial requests will need to be addressed before we can proceed with the official acceptance of your manuscript:

- Please upload all figures as individual figure files without legends. The figure legends need to be added to the ms file.
- The number of keywords needs to be reduced to 5.
- Please correct the conflict of interest subheading to "Disclosure Statement and Competing Interests"
- Please resolve this author discrepancy: Gisela Gabernet in the ms vs. Giesela Gabernet
- Please remove the author credits from the ms file. All credits need to be entered during online ms submission.
- Please correct the reference style to the EMBO reports (Harvard) style. It needs to be alphabetical, not numerical; et al needs to be used after 10 author names; DOIs should only be used for preprints and datasets that have not been published yet.
- The Appendix Figures and Tables need to be upload as one Appendix PDF file (without main ms figures). Please provide a title page for the Appendix items; the title page should have the title and all the items (figures and tables) listed with their corresponding page numbers.
- I copied the synopsis image (graphical abstract) you sent into photoshop, but it does not look very good. It would be better if you could send us this image as an image file, so jpg or tiff or similar. The image itself is good.
- The bullet points you sent are also fine. But in addition we need a short summary of your findings and their significance. 1-2 sentences max. The summary, bullet points and synopsis image will all be displayed with your ms on our website.
- "Highlights" (the bullet points) need to be removed from the ms file and upload as separate, related ms file together with the short summary.
- Please also submit a "Reagents and Tools Table" (listing key reagents, experimental models, software and relevant equipment and including their sources and relevant identifiers) with your final ms. A downloadable template (.docx) for the Reagents and Tools Table can be found in our author guidelines: <<https://www.embopress.org/page/journal/14693178/authorguide#manuscriptpreparation>>.
- Please upload the source data as one folder per ms figure.
- The manuscript sections should be in the following order: Title page - Abstract & Keywords - Introduction - Results - Discussion - Methods - Data Availability - Acknowledgments - Disclosure Statement & Competing Interests - References - Figure Legends - (Main Tables with legends if applicable) - Expanded View Figure Legends.

Please address these comments on the Figure Legends:

- Please note that the exact p values are not provided in the legends of figures 1c; 3b; 4b; 5c; 6b-c; 7a-b; 8a-b (Exact p-values need to be provided in the figure legends, as reasonable).
- Please indicate the statistical test used for data analysis in the legends of figures 2c-e.
- Please note that in figure 6c; there is a mismatch between the annotated p values in the figure legend and the annotated p values in the figure file that should be corrected.

I would like to suggest some minor changes to the abstract that needs to be written in present tense. Please let me know whether you agree with this:

Ductular reaction (DR) is the hallmark of cholestatic diseases manifested in the proliferation of bile ductules lined by biliary epithelial cells (BECs). It is commonly associated with increased risk of fibrosis and liver failure. The Receptor for Advanced Glycation End Products (RAGE) was identified as a mediator of DR during chronic injury. Yet, the direct link between RAGE-mediated DR and fibrosis as well as the mode of interaction between BECs and hepatic stellate cells (HSCs) to drive fibrosis remain elusive. Here, we delineate the specific function of RAGE on BECs during DR and its potential association with fibrosis in the context of cholestasis. Employing a biliary lineage tracing cholestatic liver injury mouse model, combined with whole transcriptome sequencing and in vitro analyses, we reveal a role for BEC-specific RAGE activity in fostering a pro-fibrotic milieu.

RAGE is predominantly expressed in BECs and contributes to DR. The Notch ligand Jagged1 is secreted from activated BECs in a RAGE-dependent manner and signals to HSCs in trans, eventually enhancing fibrosis during cholestasis.

Referee #1:

The authors addressed all my points. I am looking forward to see this interesting manuscript published.

Referee #2:

The authors have addressed most of my comments. I have no additional comments. I believe the manuscript is improved and suitable for publication.

Referee #3:

The authors have successfully answered my questions

All editorial and formatting issues were resolved by the authors.

Prof. Peter Angel
German Cancer Research Center (DKFZ)
Division of Signal Transduction and Growth Control
Im Neuenheimer Feld 280
Heidelberg
Germany

Dear Prof. Angel,

I am very pleased to accept your manuscript for publication in the next available issue of EMBO reports. Thank you for your contribution to our journal.

Yours sincerely,
